# Nutritional deficiency in an intestine-on-a-chip recapitulates injury hallmarks associated with environmental enteric dysfunction

Amir Bein[1,13,16], Cicely W. Fadel[1,2,3,16], Ben Swenor[1], Wuji Cao[1], Rani K. Powers[1,14], Diogo M. Camacho[1,15], Arash Naziripour[1], Andrew Parsons[2], Nina LoGrande[1], Sanjay Sharma[1], Seongmin Kim[1], Sasan Jalili-Firoozinezhad[1,4], Jennifer Grant[1], David T. Breault[2,5,6], Junaid Iqbal[7], Asad Ali[7], Lee A. Denson[8,9], Sean R. Moore[10], Rachelle Prantil-Baun[1], Girija Goyal[1] and Donald E. Ingber[1,11,12] ✉

Environmental enteric dysfunction (EED)—a chronic inflammatory condition of the intestine—is characterized by villus blunting, compromised intestinal barrier function and reduced nutrient absorption. Here we show that essential genotypic and phenotypic features of EED-associated intestinal injury can be reconstituted in a human intestine-on-a-chip lined by organoid-derived intestinal epithelial cells from patients with EED and cultured in nutrient-deficient medium lacking niacinamide and tryptophan. Exposure of the organ chip to such nutritional deficiencies resulted in congruent changes in six of the top ten upregulated genes that were comparable to changes seen in samples from patients with EED. Chips lined with healthy epithelium or with EED epithelium exposed to nutritional deficiencies resulted in severe villus blunting and barrier dysfunction, and in the impairment of fatty acid uptake and amino acid transport; and the chips with EED epithelium exhibited heightened secretion of inflammatory cytokines. The organ-chip model of EED-associated intestinal injury may facilitate the analysis of the molecular, genetic and nutritional bases of the disease and the testing of candidate therapeutics for it.

Environmental enteric dysfunction (EED) is a paediatric disorder characterized by chronic intestinal inflammation that is associated with malnutrition, stunted growth, cognitive impairment and attenuated response to oral vaccines[1–5]. Previously described as 'tropical enteropathy' or 'environmental enteropathy', EED has gained renewed interest in recent years due to its devastating effect on millions of children in low- and middle-income countries. The intestine of EED patients commonly exhibits villous atrophy, nutrient malabsorption, barrier dysfunction and inflammation[6,7]. As there is currently no existing in vitro model of EED and only a limited number of animal models, mechanistic understanding of this disease is limited, which has hampered efforts to define biomarkers for diagnosis or develop therapeutics[8]. For example, deficiencies of micronutrients, such as zinc[9] and vitamin A[10], may contribute to EED pathophysiology as they are associated with abnormal lactulose:mannitol (L:M) ratios—a measure of intestinal permeability. However, attempts to treat EED-related stunting by nutritional interventions have been disappointing probably due to ongoing problems with nutrient absorption and inflammation in these patients[8]. Similarly, other dietary interventions, such as administering omega-3 long-chain polyunsaturated fatty acids[11], optimizing amino acid profiles[12], supplementing with multiple micronutrients[13] or improving food digestibility through fermentation, hydrolysis or enzyme supplementation[12], have also been tested with limited success.

Inadequate dietary intake not only leads to stunting but can also increase host vulnerability to environmental factors. For example, diets low in the essential amino acid tryptophan lead to decreased antimicrobial peptide secretion and increased susceptibility to chemical-induced intestinal inflammation in mice[14]. Low serum tryptophan levels are also linked to stunting in children suffering from EED[15,16]. Tryptophan is both a building block for proteins and a precursor for niacin, melatonin and neurotransmitters (such as serotonin and tryptamine)[17], and supplementation with amino acids including tryptophan was recently shown to improve symptoms in EED patients[18]. In animal studies, symptoms of tryptophan deficiency, including anorexia and impaired growth, may occur with intakes as little as 25% below the standard requirement, which translates to 2–2.5 mg kg$^{-1}$ body weight for human infants 6–24 months old[17,19]. Recently, niacin (nicotinic acid) deficiency has also been implicated as a contributor to EED[18] and other inflammatory intestinal conditions as administration of niacin was shown to ameliorate dextran sodium sulfate-induced colitis via prostaglandin D2-mediated D prostanoid receptor 1 activation[20]. In addition,

[1]Wyss Institute for Biologically Inspired Engineering, Harvard University, Boston, MA, USA. [2]Department of Pediatrics, Harvard Medical School, Boston, MA, USA. [3]Division of Neonatology, Beth Israel Deaconess Medical Center, Boston, MA, USA. [4]Department of Bioengineering and iBB - Institute for Bioengineering and Biosciences, Instituto Superior Técnico, Universidade de Lisboa, Lisboa, Portugal. [5]Division of Endocrinology, Boston Children's Hospital, Boston, MA, USA. [6]Harvard Stem Cell Institute, Harvard University, Boston, MA, USA. [7]Department of Paediatrics and Child Health, The Aga Khan University, Karachi, Pakistan. [8]Division of Gastroenterology, Hepatology, and Nutrition, Cincinnati Children's Hospital Medical Center, Cincinnati, OH, USA. [9]Department of Pediatrics, University of Cincinnati College of Medicine, Cincinnati, OH, USA. [10]Department of Pediatrics, Division of Pediatric Gastroenterology, Hepatology, and Nutrition, University of Virginia, Charlottesville, VA, USA. [11]Harvard John A. Paulson School of Engineering and Applied Sciences, Harvard University, Cambridge, MA, USA. [12]Vascular Biology Program and Department of Surgery, Harvard Medical School and Boston Children's Hospital, Boston, MA, USA. [13]Present address: Quris Technologies, Boston, MA, USA. [14]Present address: Pluto Biosciences, Inc., Golden, CO, USA. [15]Present address: Rheos Medicines, Cambridge, MA, USA. [16]These authors contributed equally: Amir Bein, Cicely W. Fadel. ✉e-mail: don.ingber@wyss.harvard.edu

niacin serves as a precursor for coenzymes, such as nicotinamide adenine dinucleotide (NAD) and nicotinamide adenine dinucleotide phosphate (NADP), which are essential for the normal function and survival of living cells. However, a mechanistic role for malnutrition in driving EED pathophysiology in humans remains to be demonstrated.

Studying a multifactorial disease such as EED raises substantial methodical and modelling challenges and, at present, there are only a few murine models and no human in vitro models that can be used to study this disease[21]. Thus, establishing an in vitro human EED model would help to elucidate disease pathophysiology and enable the development of new prevention and therapeutic measures. Here we describe how human organ-on-a-chip microfluidic culture technology that faithfully recapitulates the structure and function of many human organs, including the intestine[22-31], can be leveraged to meet this challenge. We used human intestine chips lined with organoid-derived primary intestinal epithelium isolated from either healthy children or paediatric EED patients who were refractory to nutritional intervention. We have previously shown that these intestine chips support the formation of differentiated three-dimensional villus-like epithelial structures as well as the production of an overlying mucus layer, which require the presence of dynamic fluid flow[28,32-34]. Our studies comparing healthy versus EED intestine chips revealed that both nutritional deficiencies and genetic or epigenetic changes in the intestinal epithelium contribute to the clinically observed EED phenotype. Moreover, by comparing healthy and EED patient-derived intestine chips, we were able to study phenotypic responses to nutritional deficiencies, such as villus blunting and barrier dysfunction, which are known to be common to multiple intestinal pathologies (such as inflammatory bowel diseases and coeliac disease), and distinguish them from responses due to transcriptomic and cytokine signatures that are seen in EED.

## Results

**Nutritionally deficient EED chips recapitulate EED patient transcriptional signatures.** We previously described a two-channel microfluidic human intestine chip, lined with living human intestinal epithelium isolated from patient-derived organoids, that undergoes villus differentiation, accumulates mucus and exhibits many features of living human intestine when cultured on-chip under continuous flow with peristalsis-like mechanical deformations[28] (Fig. 1a). Additionally, transcriptional analysis demonstrated that when lined by organoid-derived duodenal epithelium, this intestine chip more closely mimicked in vivo human duodenum than the organoids used to create the chips[28]. To define the contribution of the intestinal epithelium to the EED phenotype, we created intestine chips lined with intestinal epithelial cells from organoids derived from surgical biopsies of either healthy or EED patient duodenum (healthy chips and EED chips, respectively). Compared with healthy chips, EED chips showed differential expression of 287 genes ($q < 0.05$ and fold change $\geq 1.5$, $q =$ FDR-adjusted-p-value; 86 upregulated, 201 downregulated) (Fig. 1b). EED chips showed upregulation of *MUC5AC* (previously shown to reduce inflammation, intestinal injury and bacterial translocation in an experimental intestinal injury[35]), neuregulin-4 (a known survival factor for colonic epithelium that protects against experimental intestinal injury[36,37]) and the intestinal stem cell marker *SMOC2*, whereas brush border peptidase *MME*, oxidative stress and inflammatory response controlling ectoenzyme *VNN1*, tight junction protein *CLDN10* and secreted goblet cell protein *CLCA1* were all downregulated (Supplementary Table 1).

We compared this differential gene expression profile with a recently derived clinical EED signature, which was obtained by comparing profiles of intestinal tissue samples from EED patients who were also refractory to nutritional intervention (Study of Environmental Enteropathy and Malnutrition, SEEM)[38] versus samples from healthy control patients who were investigated for gastrointestinal symptoms but had normal endoscopic and histologic findings (Cincinnati Children's Hospital Medical Center). When we compared gene profiles from EED chips versus healthy chips cultured in control medium (that is, with all nutrients present), the differentially expressed genes had some overlap with the clinical EED signature, including most notably a shared downregulation of metallothioneins (*MT1X*, *MT1A*, *MT1F*, *MT1H*, *MSMB* and *MT1M*; gene dendrogram cluster 5) (Fig. 1c).

We then carried out the same experiment but perfused both the healthy and EED intestine chips with medium deficient in niacinamide and tryptophan (−N/−T), selected on the basis of past work implicating their role in EED[15,16,19]. When we compared expression profiles from the healthy chips exposed to nutritional deficiency (healthy −N/−T chip) versus the healthy control chips, we detected differential expression of 690 genes ($q < 0.05$ and fold change $\geq 1.5$; 556 upregulated, 124 downregulated) (Fig. 1b and Supplementary Fig. 1), including upregulation of the amino acid starvation-related transcription factor *ATF4*, its downstream solute carriers (*SLC34A2*, *SLC7A5* and *SLC6A9*) and the inflammation-associated gene *LCN2* (Supplementary Table 1). Uniformity of transcriptome analysis of samples in each group was confirmed by principal component analysis (Supplementary Fig. 2). There also appeared to be a trend towards upregulation of several antimicrobial and immune response genes as seen in the clinical EED signature, but these changes did not reach statistical significance. While there was greater overlap between the transcriptome of the healthy −N/−T chip with the clinical EED transcriptome than observed with the EED control chip compared with healthy control chip, some genes were regulated in

**Fig. 1 | Nutritionally deficient EED chips recapitulate EED patient transcriptional signatures. a**, A schematic representation of small-intestine chips. **b**, Compared with healthy chips, the EED transcriptome consists of 287 differentially expressed genes (red; $q < 0.05$ and fold change $\geq 1.5$). Exposure of healthy and EED chips to -N/-T media resulted in an increased number of differentially expressed genes with 690 (yellow) and 969 genes (blue), respectively. Of these, 307 genes were differentially expressed in both healthy and EED chips exposed to nutritional deficiency (yellow and blue). Seventy-one genes were differentially expressed in EED chips when compared with healthy chips in control medium that are further affected by the addition of nutritionally deficient medium (blue and red, respectively; 3 chips for each condition). Each chip corresponds to one biological replicate. **c**, A comparison of the 20 most up- and 20 most downregulated genes from the clinical EED signature, with healthy or EED chip gene expression depicted as a heat map (red, upregulation; blue, downregulation) showing that EED -N/-T has the closest hierarchical relationship to the clinical EED signature (clinical EED Z-score distance from intestine chips results was 15.2 for the EED -N/-T vs EED; 18.7 for the healthy -N/-T vs healthy control; and 19.4 for EED control vs healthy control). The comparison for EED -N/-T to healthy control is similar and shown in Extended Data Fig. 1. Ten dendrogram gene clusters were also defined and the corresponding roots enumerated; 3 chips for each condition. Each chip corresponds to one biological replicate. **d**, Of the top 9 upregulated genes in the clinical EED signature, 6 were also upregulated when EED chips were exposed to -N/-T media; 3 chips for each condition. Each chip corresponds to one biological replicate. **e**, Functional pathway analysis was performed using the contextual language-processing programme COMPBIO. The spatial map depicts themes related to differentially expressed genes as nodes, with interconnections depicted as edges whose thickness relates to the degree of interconnectedness. The themes were also ranked according to a score representing fold enrichment over random clustering and this score determines sphere size (Supplementary Table 2); 3 chips for each condition. Each chip corresponds to one biological replicate.

an opposing direction, including upregulation of metallothioneins (Fig. 1c, gene dendrogram cluster 5).

In contrast, we observed closer unsupervised hierarchical clustering with the clinical EED signature when the EED chip was exposed to nutritional deficiency (EED −N/−T chip vs EED control chip) (Fig. 1c). Culture of the EED intestine chips in −N/−T medium yielded differential expression of 969 genes ($q < 0.05$ and fold change ≥1.5; 522 upregulated, 447 downregulated) (Fig. 1b). This

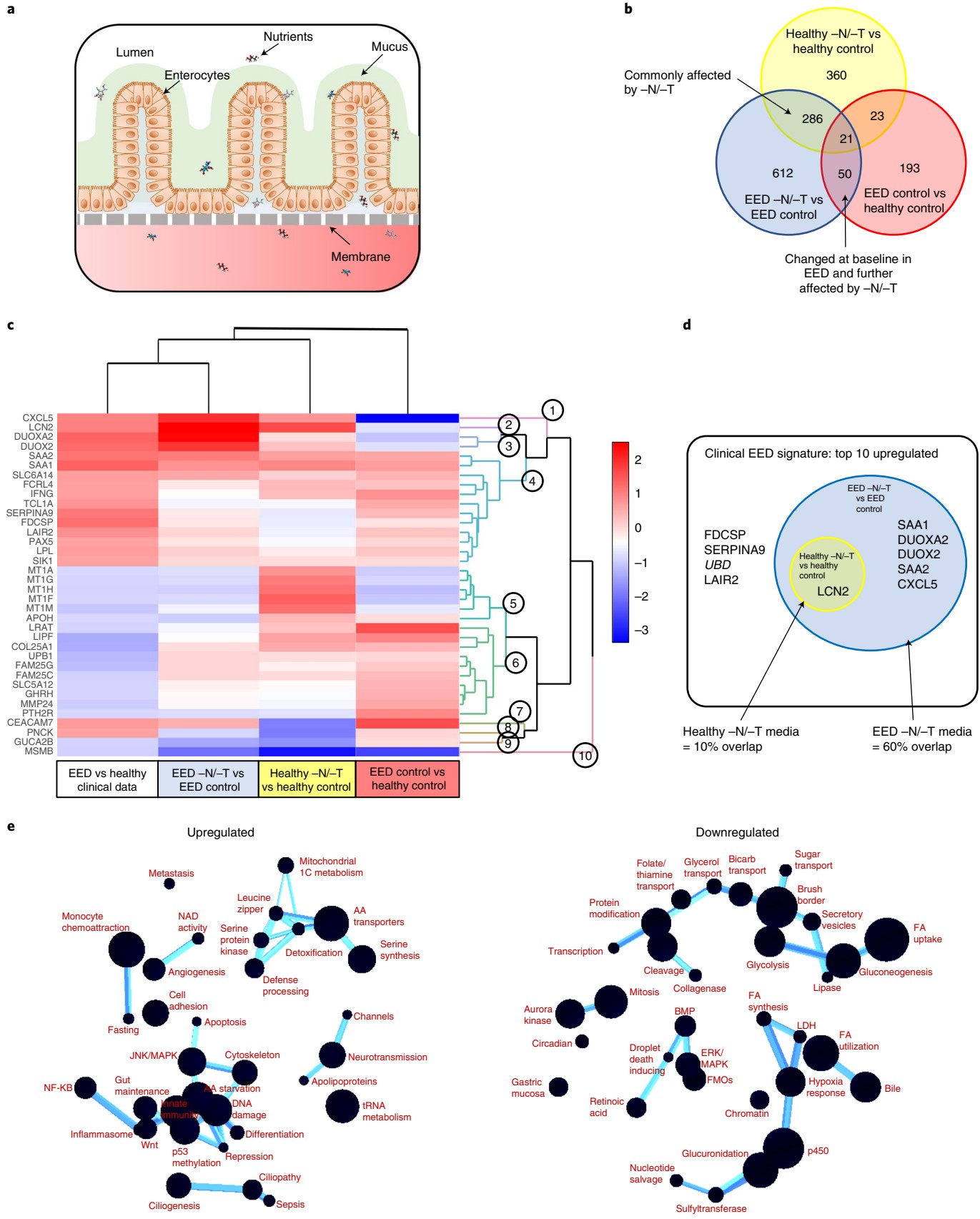

was manifested by the upregulation of antimicrobial genes (*SAA1*, *SAA2*, *DUOXA2*, *DUOX2* and *CXCL5*; gene dendrogram clusters 1, 3 and 4) and downregulation of not only metallothioneins but also metabolic and digestive genes (*SLC6A14* and *GUC2AB*; gene dendrogram clusters 4 and 9). This congruence was most noticeable among the top 10 upregulated genes of the clinical EED signature, 6 of which (60%) were also upregulated when EED chips were exposed to nutritionally deficient medium (Fig. 1d).

To identify pathways affected by exposure of EED intestine chips to −N/−T nutritional deficiency, we used a contextual language-processing programme to identify and rank functionally related clusters of genes[39]. This analysis revealed several pathways that were significantly upregulated when EED chips were exposed to −N/−T media, including a chemokine pathway (score 1,833.89, indicates fold enrichment over random association) and a pathway associated with amino acid starvation (score 1,988.09) (Fig. 1e and Supplementary Table 2). Within the amino acid starvation pathway, the *ATF4* gene is upstream of several other pathways including tRNA metabolism (score 1,547.01), DNA damage (score 1,790.28), p53 methylation (score 1,608.44) and amino acid transporters (score 2,022.97). Many of these same genes were also upregulated in nutritionally deficient healthy −N/−T chips, including *ATF4*, *TP53*, *AARS*, *YARS*, *MDM2*, *CCND2* and several amino acid transporters (Supplementary Table 1). Conversely, nutritional deficiency led to the downregulation of pathways related to fatty acid uptake (score 3,590.64), brush border structural integrity (score 3,777.69), mitosis (score 3,311.49), cytochrome p450 (score 2,581.54) and fatty acid utilization (score 2,056.86) in the EED chips. These results are consistent with the observation that the intestines of nutritionally deficient EED patients are characterized by having decreased brush border development and impaired cell growth[40–42].

We also compared gene expression changes in our model with previously identified clusters of intestinal cell-type gene markers[43–45]. In control medium, enterocyte markers were expressed at lower levels in EED chips compared with healthy chips (in particular, *MME*, *APOB* and *MTTP* were reduced by 15-fold, 2.7-fold and 2.3-fold, respectively) and were even more broadly downregulated in EED chips that were exposed to nutritional deficiency (in particular, *PCK1*, *MEP1B* and *CREB3L3* were decreased by 7.7-fold, 6.2-fold and 4.9-fold, respectively; Supplementary Fig. 3 and Table 3). Paneth cell markers were also downregulated in EED chips compared with healthy chips in control medium (for example, *MT2A*, *CFTR* and *MT1H* were suppressed by ~2–2.5-fold, respectively), but were differentially regulated when chips were exposed to nutritional deficiency. Healthy chips exposed to nutritional deficiency upregulated Paneth cell markers (for example, *MT1H*, *MT1M* and *MT1G* all increased by ~2-fold), while EED chips exposed to nutritional deficiency exhibited further Paneth cell marker downregulation (specifically, *ID1*, *SULT1E1* and *MT1X* were reduced ~2–2.5-fold). These findings are consistent with the histopathological finding of Paneth cell depletion in EED patient intestinal biopsies[38].

**Intestinal villus atrophy and barrier compromise.** As the transcriptomic analysis revealed downregulation in pathways involved in cell growth and intestinal barrier formation, we carried out differential interference contrast (DIC) and immunofluorescence microscopic analyses, which indeed confirmed that both healthy and EED intestine chips showed dramatically reduced growth of villus-like structures when cultured under nutrition deficient (−N/−T) conditions compared with healthy and EED control chips perfused with complete medium (Fig. 2a,b). Quantification of the height of the epithelium revealed that removal of these nutrients resulted in significant villus blunting in both healthy and EED chips in response to −N/−T deficiency, as indicated by a 70% and 80% reduction in epithelial height, respectively, compared with the same chips cultured in complete medium (Fig. 2c).

Another gene cluster identified as being preferentially sensitive to nutritional deficiency includes genes governing brush border structural integrity. These genes included myosin 1a (*MYO1A*), which links actin to the overlying apical membrane and whose absence results in irregularities of microvilli packing and length[46]; protocadherin-24 (*CDHR2*) that forms links between adjacent microvilli and is the target of enterohemorrhagic *Escherichia coli*-mediated brush border damage[47]; and mucin-like protocadherin (*CDHR5*) that forms heterophilic complexes with *CDHR2* (Supplementary Fig. 4). Indeed, scanning electron microscopic imaging revealed that culturing healthy intestinal epithelium in −N/−T medium on-chip resulted in severe loss of apical microvilli (as well as links between adjacent microvilli) relative to control enterocytes that had their entire surface covered with tightly packed microvilli (Supplementary Fig. 5). Analysis of mucus accumulation by live imaging and staining with fluorescent lectin also revealed that both the healthy and EED chips exhibited a much thinner mucus layer when exposed to nutrient-deficient conditions (Fig. 2d,e).

Following these observations of structural changes due to exposure to nutritional deficiency, we next leveraged the advantage of using a two-channel microfluidic intestine chip (Fig. 1a) (that is, as opposed to intestinal organoids cultured within a solid extracellular matrix gel) to assess the effect of exposure to −N/−T medium on differences in intestinal barrier function between healthy and EED chips. We compared apparent permeability ($P_{app}$) values, which were measured by calculating the transfer of Cascade Blue fluorescent tracer (596 Da) from the epithelial lumen in the top channel to the underlying parallel channel below. Both healthy and EED control chips exhibited a tight barrier under baseline conditions (within the $10^{-7}$ $P_{app}$ range) and displayed small but statistically significant reductions in barrier function when exposed to nutritional deficiency, as indicated by 8.9- and 2.5-fold increases in $P_{app}$ for the healthy and EED chips, respectively (Fig. 2f).

**Reduced nutrient absorption.** Our transcriptomic analysis also revealed that nutritional deficiency resulted in the downregulation of multiple genes associated with the uptake and processing of important nutritional components, including fatty acids, certain amino acids and carbohydrates (Figs. 1e, 3a, 4a and Extended Data Fig. 2). This is clinically relevant because reduced absorption of nutrients is another hallmark of EED, and it affects weight and linear growth as well as cognitive development in children[48–51]. For example, expression levels for fatty acid translocase, cluster of differentiation 36 (*CD36*), microsomal triglyceride transfer protein (*MTTP*), apolipoprotein B (*ApoB*) and apolipoprotein C-III (*ApoC3*) were all lower in nutritionally deficient epithelium (Fig. 3a). Similarly, when we used immunofluorescence microscopy to assess expression of ApoB protein, which is a marker of chylomicron and fat metabolism in the intestine[52], we found that exposure to −N/−T nutritional deficiency resulted in significant downregulation of *ApoB* in EED intestine chips (Fig. 3b and Supplementary Fig. 6). Furthermore, when we quantified cellular uptake of fatty acids using fluorescently labelled dodecanoic acid, we found that exposure to nutritional deficiency reduced fatty acid uptake by 1.68- and 1.69-fold in healthy −N/−T chips and EED −N/−T chips, respectively, compared with healthy and EED control chips (Fig. 3c). These findings are consistent with clinical data that similarly show impaired fatty acid metabolism in children suffering from EED[16] and suggest that nutritional deficiency alone is sufficient to reduce fatty acid uptake even in healthy intestine.

**Abnormal amino acid uptake and metabolism.** Children suffering from EED exhibit impaired development, and protein availability from the diet is a key factor responsible for linear growth; thus, we next explored differences in uptake by healthy and EED

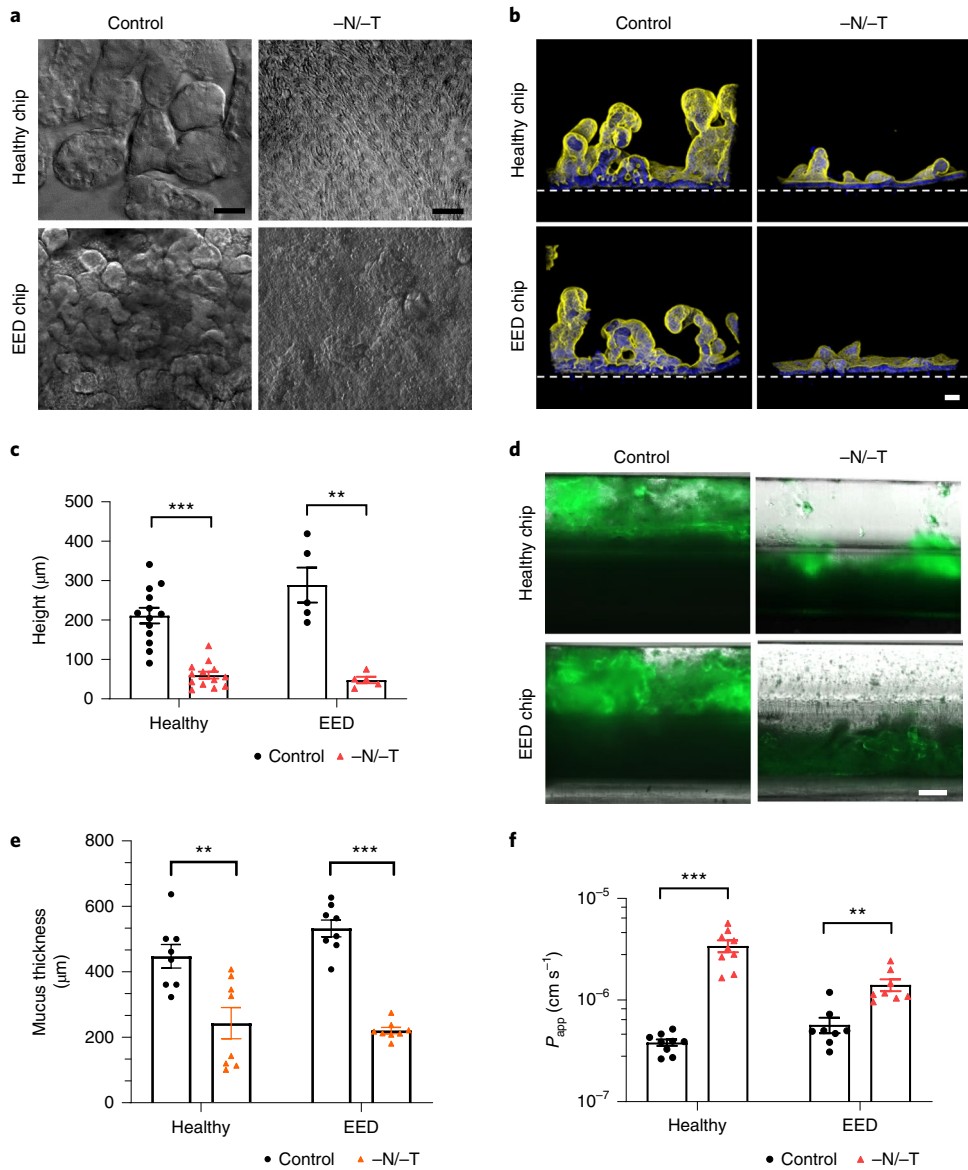

**Fig. 2 | Intestinal villus atrophy and barrier compromise. a**, DIC imaging of intestine chips, top-down view (scale bar, 50 μm). **b**, Immunofluorescence cross-section micrographs showing villus-like structures in the intestine chips. Yellow, phalloidin; Blue, Hoechst; Dashed white line, upper surface of chip membrane. Scale bar, 50 μm. **c**, Villus-like structures height differences between control and −N/−T, healthy (mean: 211 μm for control and 60 μm for −N/−T) and EED (mean: 288 μm for control and 48 μm for −N/−T) intestine chips. Healthy, $P < 0.000001$, EED, $P = 0.000685$, multiple Student's $t$-test. Each datapoint on the graph represents the average of 4 to 5 measurement points (membrane to top of the villi) measured in 2 to 3 cross-sectional images of the chip in at least 2 chips per condition. Two different healthy donors and one EED donor were used. Each chip corresponds to one biological replicate. **d**, Immunofluorescence microscopic views of longitudinal section through the two-channel intestine chips showing the mucus layer overlying the epithelium in the upper channel stained with Alexa 488-conjugated lectin. Scale bar, 250 μm. **e**, Graph showing differences in mucus thickness between control and −N/−T, healthy (mean: 448 μm for control and 243 μm for −N/−T) and EED (mean: 533 μm for control and 221 μm for −N/−T) intestine chips. Each datapoint on the graph represents the average of 2 measurement points measured in views similar to those shown in **d**; 2 chips per condition with chips created from cells from two different healthy donors and one EED donor. Healthy, $P = 0.004092$; EED, $P < 0.000001$, multiple Student's $t$-test. Each chip corresponds to one biological replicate. **f**, Graph showing differences in $P_{app}$ between control and −N/−T, healthy (mean: $3.82 \times 10^{-7}$ for control and $3.41 \times 10^{-6}$ for −N/−T) and EED (mean: $5.71 \times 10^{-7}$ for control and $1.41 \times 10^{-6}$ for −N/−T) intestine chips (9 for healthy chips and 8 for EED intestine chips). Healthy, $P = 0.000005$; EED, $P = 0.001382$, multiple Student's $t$-test. Each chip corresponds to one biological replicate. Graphed data in **c**, **e** and **f** are mean ± s.e.m.

intestine chips. In the intestine, dietary protein is broken down into short peptides and free amino acids that are taken up by enterocytes, these short peptides and free amino acids serving as building blocks and energy sources for various organs and tissues[48]. Transcriptomic analysis revealed absorption and metabolizing factors that were downregulated in EED control chips vs healthy control chips, including the solute carrier family 2 (facilitated glucose transporter) member 2 (*SLC2A2*) and solute carrier family 2 (facilitated glucose/fructose transporter) member 5 (*SLC2A5*). Other nutrient transporters (such as amino acid absorption and metabolizing factors) that were found to be significantly downregulated in EED chips in response to −N/−T deficiency (EED

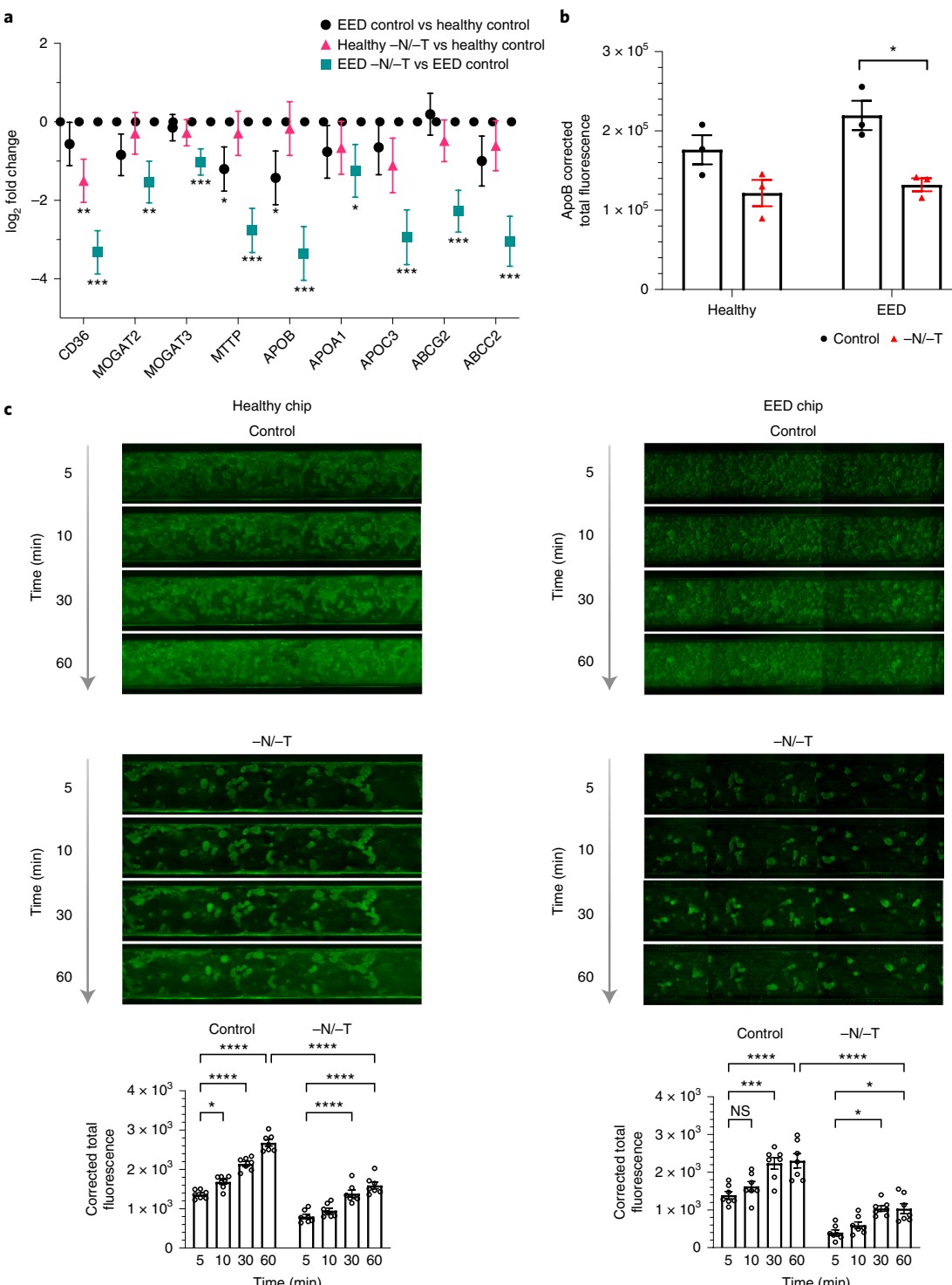

**Fig. 3 | Reduced nutrient absorption. a**, Transcriptional pathway analysis revealed a strong theme of downregulation for genes related to fatty acid uptake when EED chips were exposed to −N/−T media. The $\log_2$ fold changes between the average expression levels are depicted along with the associated 95% confidence interval. These changes included a 9.8-fold downregulation of the receptor *CD36* ($q < 0.001$), a 10.2-fold downregulation of *ApoB* ($q < .001$) and an 8.2-fold downregulation of *ABCC2* ($q < 0.001$). There was a similar trend towards downregulation when healthy chips were exposed to −N/−T media, but most gene changes did not reach statistical significance; 3 chips for each condition. Unmoderated Student's *t*-test and the FDR method for multiple testing correction. Each chip corresponds to one biological replicate. **b**, Differences in *ApoB* corrected total fluorescence expression in control and −N/−T healthy and EED intestine chips. Healthy (mean: $1.7 \times 10^5$ for control and $1.2 \times 10^5$ for −N/−T), not significant; EED (mean: $2.2 \times 10^5$ for control and $1.3 \times 10^5$ for −N/−T), $P = 0.012284$. Each datapoint on the graph represents the average of corrected total fluorescence expression from 3 different measurement areas of intestine chip cross-sections, from at least 2 chips per condition. Each chip corresponds to one biological replicate. **c**, Differences in fluorescently labelled dodecanoic fatty acid uptake by healthy (left) and EED (right) intestine chips at the 5 min, 10 min, 30 min and 60 min timepoints. For each timepoint, 7 images covering the entire area of a representative chip from each condition were used to calculate the corrected total fluorescence expression (bottom). Graphed data in **b** and **c** are mean ± s.e.m. \**P* < 0.04, \*\*\**P* ≤ 0.0008, \*\*\*\**P* < 0.0001 by a 2-way ANOVA.

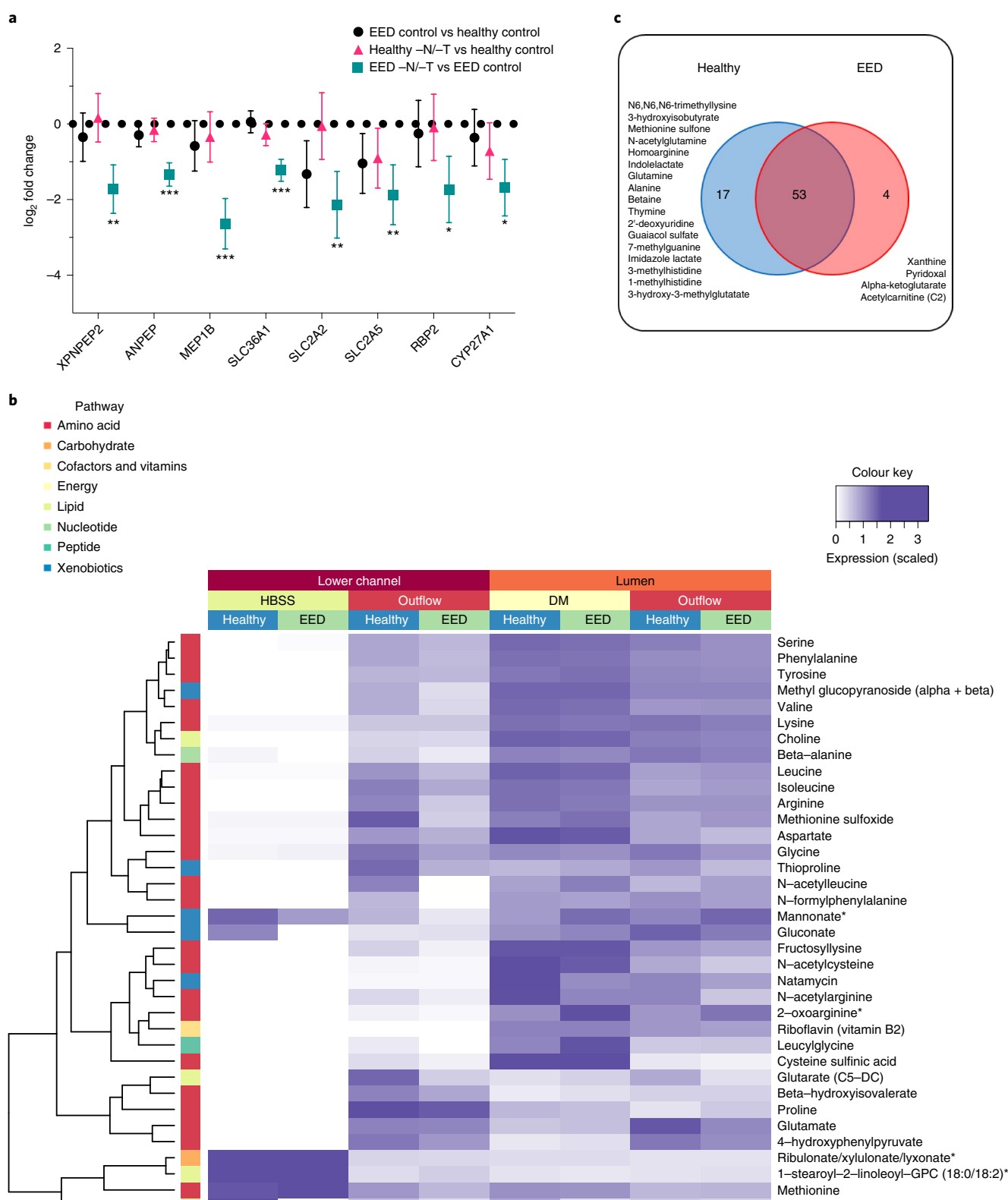

**Fig. 4 | Abnormal amino acid uptake and metabolism. a**, Amino acid processing and transport was among the strongly downregulated themes when EED chips were exposed to −N/−T media. This included a 6.2-fold downregulation of *MEP1B* ($q < 0.001$), a 4.4-fold downregulation of *SLC2A2* ($q = 0.0074$) and a 3.3-fold downregulation of *XPNPEP2* ($q = 0.0036$), which are shown as the log$_2$ fold changes between the average expression levels along with the associated 95% confidence interval. There was occasional downregulation of these genes when healthy chips were exposed to −N/−T media, but none that achieved statistical significance; 3 chips for each condition. Each chip corresponds to one biological replicate. **b**, Heat map showing 36 metabolites taken up and transferred from the luminal medium to the lower channel of the intestine chips at higher abundance in the healthy vs EED chips (DM, differentiation medium; 3 chips for each condition). Each chip corresponds to one biological replicate. **c**, Venn diagram showing the number of common and unique metabolites secreted by the cells in intestine chips.

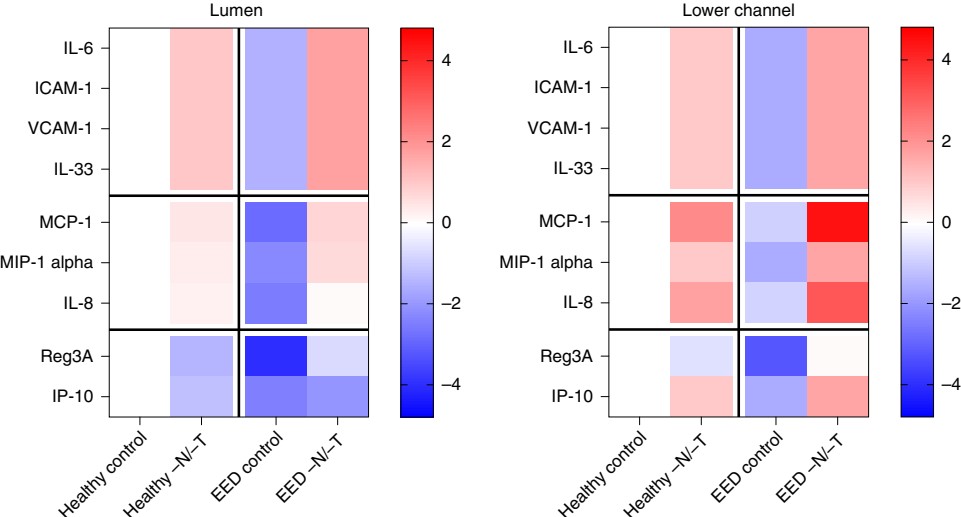

**Fig. 5 | Altered inflammatory mediators.** Heat maps showing differential expression of 9 cytokines secreted into the lumen or lower channel of the control or −N/−T, healthy and EED intestine chips and quantified using a bead-based multiplexed ELISA. The colour-coded scale represents the log$_2$ fold change in expression; 3–6 chips for each condition. Each chip corresponds to one biological replicate.

−N/−T vs EED control), include *SLC36A1*, which encodes the proton-coupled amino acid transporter 1, *ANPEP* membrane enzyme alanyl aminopeptidase that is responsible for peptide digestion at the brush border, retinol binding protein 2 (*RBP2*), cytochrome P450, family 27, subfamily A, and polypeptide 1 (*CYP27A1*) (Fig. 4a and Supplementary Fig. 7).

We then assessed differences in absorption of nutrients between the healthy and EED intestine chips grown in control medium by performing untargeted metabolomic analysis by liquid chromatography tandem mass spectrometry (LC–MS/MS) to analyse epithelial uptake of nutrients and transfer of these molecules from the lumen of the intestinal epithelium in the top channel to the underlying basal channel (Fig. 1a). We detected 36 metabolites (out of >500 identified) that exhibited lower transport in EED intestine chips compared with healthy chips grown in control medium. These included mainly amino acid metabolites, but also metabolites related to nucleotides, cofactors, lipids, carbohydrates and xenobiotic pathways (Fig. 4b; an extended list of metabolites analysed can be found in Supplementary Fig. 8). Interestingly, 9 of these metabolites were amino acids previously identified as being reduced in serum of Malawian stunted children[53]. These included essential amino acids (isoleucine, leucine, methionine, phenylalanine, lysine), conditionally essential amino acids (arginine, glycine) and non-essential amino acids (glutamate, serine).

Our LC–MS/MS analysis also revealed 74 metabolites that were secreted by the intestinal cells as they were absent or at extremely low levels (<5%) in the perfusion medium. Of these metabolites, 17 were unique to the healthy control chips and included products of pathways related to metabolism of amino acids (N6,N6,N6-trimethyllysine, 3-hydroxyisobutyrate, methionine sulfone, *N*-acetylglutamine, homoarginine, indolelactate, glutamine, alanine, betaine, imidazole lactate, 3-methylhistidine, 1-methylhistidine), nucleotides (thymine, 2'-deoxyuridine, 7-methylguanine), xenobiotic metabolism (guaiacol sulfate) or lipids (3-hydroxy-3-methylglutarate). Interestingly, we also identified 4 metabolites unique to the EED chips cultured in control medium, including products of purine nucleotide metabolism (xanthine), vitamin B6 metabolism (pyridoxal), citric acid cycle/energy metabolism (alpha-ketoglutarate) and fatty acid metabolism (acetylcarnitine (C2)) (Fig. 4c).

To assess whether the observed differential uptake and metabolism of molecules in the intestine chips were transporter dependent, we quantified uptake of the dipeptide, glycyl-sarcosine (Gly-Sar), by the epithelial cells in the top channel, and its transfer to the lower channel using LC–MS/MS. This analysis revealed reduced uptake and transport of Gly-Sar in nutritionally deficient healthy intestine chips compared with the same chips cultured in control medium (Supplementary Fig. 9). In addition, these studies confirmed that these effects were due to transport through the *PEPT1* transporter, and not due to passive inter- or intracellular diffusion, as this response could be completely prevented by adding Gly-Gly dipeptide (1 mM), which is a specific inhibitor of this transporter[54] (Supplementary Fig. 9).

**Altered inflammatory mediators.** Altered intestinal inflammation is a key component of EED and our transcriptional analysis revealed that genes encoding key inflammatory marker proteins, such as lipocalin 2 (*LCN2*) and regenerating islet-derived protein 3 alpha (*REG3A*), were upregulated when EED or healthy chips were exposed to nutritional deficiency (Fig. 1c and Extended Data Fig. 3). Indeed, when we quantified the expression of nine key intestinal cytokines using a bead-based multiplexed ELISA assay, we detected higher levels of several cytokines (*IL-6, ICAM, VCAM, IL-33, MCP-1, MIP-1 alpha* and *IL-8*) in the epithelial lumen of healthy −N/−T intestine chips compared with the same chips cultured in control medium (Fig. 5a). The antimicrobial peptide *Reg3A* was also downregulated by nutritional deficiency in these chips. Interestingly, EED intestine chips grown in complete medium displayed reduced levels of all of the secreted cytokines analysed, while their levels were significantly upregulated when these chips were grown in nutritionally deficient medium (Fig. 5a). Similar responses were observed when we analysed cytokine levels in the lower parenchymal or vascular channel; however, significantly higher levels of the inflammatory cytokines *IL-8, IP-10* and *MCP-1* were observed (Fig. 5a). Interestingly, we found that intestine chips that were not exposed to mechanical peristalsis-like deformation secreted lower levels of *IL-8* and *MCP-1* (Extended Data Fig. 4).

## Discussion
Given the importance of studying the pathophysiology and the underlying mechanisms of EED, and the current lack of human

in vitro models, this study leveraged the human intestine chip technology to create an in vitro EED model using cells obtained from EED and healthy patient intestinal biopsies. The intestine chip offers a unique advantage over other advanced in vitro models, such as intestinal organoids, because direct access to the two parallel flow channels of the device enable quantitation of intestinal barrier function as well as transepithelial absorption and transport that are not possible in static three-dimensional enteroid cultures. Using this approach, we explored the effect of nutritional deficiencies on the manifestation of the disease at both the phenotypic and functional levels. Importantly, our results showed that human intestine chips lined by intestinal epithelial cells isolated from organoids derived from EED patients mimic key features of the transcriptome signature of EED patients, including upregulation of antimicrobial genes and downregulation of metallothioneins and genes involved in digestion and metabolism, but only when exposed to nutritional deficiency, which we modelled by removing niacin and tryptophan from the medium. In contrast, nutritional deficiency induced similar intestinal villus atrophy, disruption of barrier function, and changes in amino acid and fatty acid absorption in chips lined by cells from both healthy and EED patients. Thus, we were able to attribute these various responses specifically to nutritional deficiency, genetic or epigenetic changes in the intestinal epithelium, or a combination of both—distinctions that have not been possible to make in clinical studies, at least for the clinically interesting subpopulation of EED patients studied here who are referred to biopsy after failing nutritional intervention.

Importantly, although the number of EED donors used in this study was limited, the results obtained with EED intestine chips were consistent between donors and compared favourably with genetic data from clinical studies with much larger numbers of EED patients carried out in Pakistan (SEEM study), as well as in similar clinical studies in Zambia and Bangladesh. The Zambian study compared stunted paediatric patients who were unresponsive to nutritional intervention, to paediatric patients or adults with severe acute malnutrition, and EED relevant changes were seen in NADPH oxidases (such as *DUOXA2*), chemokines (such as *CXCL5*), metallothioneins (for example, *MT1X*), membrane ion transport (such as *SLC4A7* and *SLC23A3*), antimicrobial defence (in particular, *SAA1*, *SAAA2* and *DUOXA2*) and mucosal protection (such as *TFF1*, *TFF2* and *MUC6*)[55]. Interestingly, a duodenal proteomic profile that correlated with dysbiosis in a nutritionally refractory stunted Bangladeshi patient cohort also showed overlap with genes related to brush border structure, mucosal function and inflammation (downregulation of *CDHR5*, *CLIC5* and *TFF4*; upregulation of *LCN2*)[56]. Additional organ-on-a-chip studies will need to be carried out with cells from a larger number of EED patients to rule out any potential biases due to small sample size and to determine their generalizability to other EED patient subpopulations.

Owing to the modular design of the intestine chip, future studies using the EED intestine chip could provide a more in-depth analysis of the immune compartment in EED by incorporating immune cells into the model. Because the immune response plays an important role in EED, addition of this complexity to the model could provide greater insight into associated inflammatory processes, their interplay with nutritional deficiencies as well as bacterial infections, and most importantly, provide an improved testbed for EED treatments. As these intestine chips can support co-culture with complex human gut microbiome[26], this model could also enable future studies with and without healthy or EED patient-derived duodenal bacterial isolates to further define the separate and combinatorial causes of dysbiosis and intestinal dysfunction in EED pathophysiology.

Malnutrition in EED can be regarded as both a cause and an outcome of the disease. In this study, we examined the effect of dual deficiency of an essential amino acid tryptophan and the vitamin niacin (in the form of niacinamide) because of their reported

effect on intestinal development and function, and the suggested correlation between their deficiencies and EED development[18,53]. Remarkably, we observed a 6-fold increase in the number of genes differentially expressed in EED intestine chips exposed to nutritional deficiency compared with healthy chips grown in complete medium. Moreover, many of the affected pathways were associated with nutritional uptake and cellular energy processing. These results imply that nutritional deficiency itself leads to derangements in nutritional processing that create a positive feedback loop, which further worsens the nutritional deficiency in EED patients.

Villus blunting and barrier dysfunction are hallmarks not only of EED, but also of other intestinal pathological conditions, such as inflammatory bowel diseases, coeliac, diarrhoea and small-intestine bacterial overgrowth[57,58]. Indeed, we were able to show in this study that these phenotypes can manifest because of nutritional deficiencies regardless of whether the intestinal epithelium was derived from healthy or EED patients. This implies that villus blunting and barrier dysfunction observed in EED may be a response to environmental conditions, rather than an inherent genetic or developmental feature of this disease. This finding has high clinical relevance because the intestinal epithelium undergoes continuous shedding and renewal every 3–5 d; hence, the chronic negative effect of nutritional deficiencies might explain a fundamental aspect of the malabsorption and poor response to oral vaccination seen in EED patients. The ability to dissect and assess the factors leading to villus blunting in EED (as well as in other intestinal conditions) in vitro is a unique capability enabled by intestine chips that support the formation of three-dimensional villus-like structures as well as mucus production and quantification of transepithelial transport, absorption and secretion.

Fat is a macronutrient responsible for 30–40% of total caloric intake in children and 20–35% in adults, and fatty acid composition has a direct effect on health and development, including inflammatory status and cognitive development[50,51]. Our transcriptomic analysis was sensitive enough to detect several genes, including *ApoB*, that were downregulated in nutritionally deficient epithelium. We confirmed that the expression of this molecule is decreased, using immunofluorescence microscopy, and that fatty acid absorption is impaired in both EED −N/−T and healthy −N/−T chips (compared with their respective controls), using a fatty acid uptake assay.

Given that dietary protein is an important macronutrient directly linked to linear growth and the results from our transcriptome-wide analysis revealed downregulation of amino acid transporters in EED chips, we also conducted an untargeted metabolomic analysis that identified several amino acids that are significantly reduced in the EED control chips compared with healthy control chips. More interestingly, we identified metabolites that were secreted by the intestinal epithelium into the lower channel. Thus, the intestine chip may be used as a nutritional and metabolic screening tool where uptake, utilization and secretion of specific metabolites by and through the intestinal epithelium can be followed in high resolution and quantified over time. Moreover, future analysis of molecules released into the lower flow channel could lead to identification of biomarkers of disease severity and/or progression that might be detectable in blood.

While reduction of intestinal absorptive surface area due to villus blunting caused by nutritional deficiency could impair nutrient uptake by the intestine, exposure to nutritional deficiency also directly suppressed expression of multiple genes related to nutrient absorption, specifically in chips lined by cells from EED patients. Thus, these results suggest that nutritional deficiency has a two-fold effect in these patients, which would probably manifest in a greater degree of intestinal dysfunction and a more severe EED phenotype. In our study, we also found that EED intestine chips exposed to nutritional deficiencies produced greater amounts of inflammatory cytokines compared with healthy chips grown under the same

−N/−T conditions. This is a critical detail as inflamed intestine has higher caloric demands for basic maintenance and renewal, which could result in a negative caloric balance unable to support catch-up growth. Furthermore, chronic inflammation may negatively affect the efficacy of oral vaccines in the EED intestine. As such, it is possible that the lack of catch-up growth in EED children receiving nutritional intervention is due to current interventions only supplying nutrients aimed at replenishing the deficiency in tissues responsible for linear growth (bone, muscles)[59]. A more effective approach might be to first administer a diet composition that preferentially promotes intestinal recovery with formation of new villi and restores increased intestinal absorptive area before moving to supplementation required for catch-up growth.

The ability of the intestine chips to allow the study of the effects of environmental factors (such as cell source and nutrient levels) individually or in combination in a controlled manner and to explore multiple clinically relevant outcomes, such as cell and tissue morphology, barrier function, nutrient metabolism and absorption, inflammatory status and transcriptome modifications, enabled us to distinguish between manifestations more common to multiple intestinal diseases versus responses that are unique to EED. For example, we identified the CD36 gene, which has several functions (in particular, as a fatty acid translocase, as a regulator of inflammation, and in oxidative stress and angiogenesis) relevant to EED as well as to intestinal cancer and other intestinal diseases, to be significantly downregulated under nutritional deficiencies in both healthy and EED intestine chips. Remarkably, our findings relating to EED are directly in line with recently published clinical data that explored unique signatures of EED-affected children compared with healthy controls and children with coeliac disease[38]. Thus, this in vitro EED model may be useful for gaining further insight into the pathophysiology of this disease as well as for development of potent therapeutics. The intestine chip could also find uses in personalized medicine and nutrition by leveraging clinical biopsies, potentially allowing for personalized (patient-specific) digestion, absorption and allergic reactions to be assessed for different nutrients.

## Methods

**Organoid cultures and intestine chips.** Organoids from healthy donors or EED patients were generated from biopsy samples collected during exploratory gastroscopy following a procedure previously described[60]. A total of 3 healthy and 2 EED donors were used to generate the data in this study (Supplementary Table 4). For the healthy chips, de-identified endoscopic tissue biopsies were collected at Boston Children's Hospital from grossly unaffected (macroscopically normal) areas of the duodenum in 2-, 10- and 11-year-old patients undergoing endoscopy for gastrointestinal complaints. Informed consent was obtained at Boston Children's Hospital from the donors' guardians. All methods were carried out in accordance with the approval of the Institutional Review Board of Boston Children's Hospital (Protocol number IRB-P00000529). For the EED chips, samples were collected in Pakistan and newborns in this study were registered and followed every month for anthropometry. Children with weight for height (WHZ) scores of less than −2 at their 3–6 month follow up visit were selected as cases. The de-identified endoscopic tissue biopsies were collected from affected areas of the duodenum in 1.5-year-old and 1.9-year-old patients undergoing endoscopy following unsuccessful educational and nutritional intervention for wasting. Informed consent was obtained at the household level in a rural district of Matiari, Sind, Pakistan from the donors' guardians. All methods were carried out in accordance with the AKU Ethical Review Committee's approval (ERC number 3836-Ped-ERC-15). Organoids were kept in complete growth medium[28,60], and passaged every 7 d in a 1:4 ratio. Before cell seeding, S-1 chips (Emulate) were activated using ER1/2 (Emulate) and UV exposure for 20 min. Chips were then coated with 200 μg ml⁻¹ collagen I (BD Corning) and 100 μg ml⁻¹ Matrigel (BD Corning) in serum-free DMEM-F12 (Gibco) for 2 h at 37 °C. After washing, organoids were broken into smaller fragments using enzymatic activity (TrypLE, Gibco) and seeded in the luminal upper channel of the chips. They were then allowed to adhere for 24 h before introduction of flow and mechanical deformation as previously described[28]. For the −N/−T treatment, niacinamide and tryptophan were removed from the basal medium (DMEM-F12, Gibco) used to prepare the expansion culture medium (used for the luminal and lower channels) and no additional niacinamide was added[28]. After 16–18 d in culture with continuous flow (60 μl h⁻¹) and mechanical deformation (10%, 0.15 Hz), the medium was changed to differentiation medium (serum and Wnt-3A free[28], and −N/−T free for the respective group) in the luminal top channel and expansion culture medium in the lower channel for 4 additional days.

**Microarray sequencing and bioinformatics analysis.** Initial microarray experiments were carried out using one healthy donor ($n = 3$ biological replicates) and one EED donor ($n = 3$ biological replicates) and were reflective of a recent clinical EED transcriptomic signature derived from a larger population (SEEM study, $n = 25$ healthy donors, 52 EED donors). Subsequent validation studies were carried out using 1–3 healthy donors ($n = 3–9$ biological replicates) and 1–2 EED donors ($n = 3–8$ biological replicates) per experiment. RNA samples were processed using the Genechip WT PLUS reagent kit and hybridized to Affymetrix Human Clariom D arrays. Robust multichip average (RMA) was used to generate normalized expression intensity data for subsequent analysis (R package, oligo; function, rma). Differential expression analysis was performed with limma[61] R package for each comparison pair; gene expression values were averaged for each condition. Template matching was used to extract genes that are differentially expressed between these conditions. Differential gene expression heat map analysis was performed using Euclidean distance and McQuitty's linkage within the R package heatmaply[39]. To normalize values across experiments, columns were scaled to generate a $Z$-score. Pathway analysis was performed using the natural language-processing algorithm COmprehensive Multi-omics Platform for Biological InterpretatiON (COMPBIO) to generate a holistic contextual map of the core biological themes associated with gene expression changes. Enriched concepts associated with differentially expressed genes were compiled from PubMed abstracts using contextual language-processing. Themes were scored using a complex function that incorporates an empirical $P$ value and a standard score similar to a $Z$-score to give a final value representing fold enrichment over a random clustering as follows: 3–9, weak relationship; 10–99, modest relationship; 100–999, strong relationship; 1,000+, very strong relationship. A theme map was generated where themes are represented as nodes and interconnections between themes are represented as edges, with the thickness of an edge relating to the degree of interconnection.

We also compared intestine chip results with those obtained from samples from the SEEM study in which Pakistani patients were enrolled as newborns and their growth trajectories were followed to 24 months. Children who were malnourished (weight for height < −2.0) at 3–6 months of age were selected and given nutritional intervention if they continued to be malnourished at 9 months of age. Those EED patients who were unresponsive to nutritional intervention underwent esophagogastroduodenoscopy and histological evaluation as part of the clinical workup. SEEM samples were sequenced using RNA-seq technology as previously detailed[38]. Comparisons between the two studies were made using $Z$-score-scaled fold-change data.

**Immunofluorescence microscopy.** Immunofluorescence microscopic imaging was carried out using the following steps: the apical and basal channels of the chips were gently washed with PBS and fixed with 4% paraformaldehyde (Electron Microscopy Sciences, 157-4) in PBS for 30 min, then washed twice with PBS and kept at 4 °C. The fixed samples were sectioned to 150–250 μm sections using a vibratome (Leica), and then permeabilized and blocked with 0.1% Triton X-100 solution and 10% donkey serum in PBS for 30 min at room temperature. Then primary antibody (Apo-B; Abcam, ab20737) was added (1:100 in 1.5% BSA/PBS solution) and the samples were incubated overnight at 4 °C, followed by multiple PBS washes. Cells were then incubated with secondary fluorescent antibody (Invitrogen, SA5-10038) and phalloidin (Invitrogen, A12380) at room temperature for 60 min and washed with PBS; nuclei were co-stained with Hoechst 33342 (Sigma, 14533). Microscopy was performed with a laser scanning confocal microscope (Leica SP5 X MP DMI-6000 or Zeiss TIRF/LSM 710).

**Paracellular permeability measurements.** To assess paracellular permeability, 50 μg ml⁻¹ of Cascade blue (596 Da; ThermoFisher, C687) was introduced to the luminal channel (at 60 ml h⁻¹). After flowing overnight to saturate the microfluidic channels, outflows were discarded and collection for measurements were conducted for ~24 h. The fluorescence intensities (390 nm/420 nm) of the top and bottom channel effluents were measured using a multimode plate reader (BioTek NEO). The apical-to-basolateral flux of the paracellular marker was calculated using the following equation: $P_{app} = (dQ/dt)/AdC$, where $dQ/dt$ (g s⁻¹) is molecular flux, $A$ (cm²) is the total area of diffusion and $dC$ (mg ml⁻¹) is the average gradient.

**Mucus assessment.** Mucus was visualized using a wheat germ agglutinin (WGA)–Alexa Fluor 488 conjugate (ThermoFisher, W11261) for live cell imaging, as described previously[26] with some modifications. Briefly, WGA solution (25 μg ml⁻¹ in Hanks' Balanced Salt Solution, HBSS) was allowed to flow through the epithelium channel for 30 min and then washed with continuous flow of HBSS for 30 min. Intestine chips were then cut sideways parallel to the length of the channel and imaged with an epifluorescence microscope (Zeiss Axio Observer Z1) with ×5 objective.

**Fatty acid uptake.** Chips were starved for 1 h by replacing the luminal and lower channel media with HBSS. Then, fluorescently labelled dodecanoic acid combined with a quencher (to eliminate any unspecific signal) were added according to

1245

the manufacturer instructions (BioVision, K408) to the luminal upper channel of the intestine chips. The entire length of the channels was then imaged with an epifluorescence microscope (Excitation/Emission = 488/523 nm, Zeiss Axio Observer Z1) with ×5 objective at 5, 10, 30 and 60 min.

**Metabolomics.** The medium in the lower channel of the chips was changed to HBSS and allowed to flow for 30 min to clear any residues. Then collection of outflows was conducted over 5 h. Samples were frozen immediately after collection (−80 °C) and submitted for LC–MS/MS analysis. The Metabolon global metabolomics platform was used to measure biochemicals in cell and media samples. Samples were prepared using the automated MicroLab STAR system (Hamilton). Several recovery standards were added before the first step in the extraction process for quality control purposes. To remove protein, dissociate small molecules bound to protein or trapped in the precipitated protein matrix, and to recover chemically diverse metabolites, proteins were precipitated with methanol under vigorous shaking for 2 min (Glen Mills GenoGrinder 2000), followed by centrifugation. The resulting extract was divided into five fractions: two for analysis by two separate reverse phase (RP)/Ultra Performance Liquid Chromatography (UPLC)–MS/MS methods with positive ion mode electrospray ionization (ESI), one for analysis by RP/UPLC–MS/MS with negative ion mode ESI, one for analysis by Hydrophilic interaction (HILIC)/UPLC–MS/MS with negative ion mode ESI, and one sample was reserved for backup. Samples were placed briefly on a TurboVap (Zymark) to remove the organic solvent. Raw data were extracted, peak-identified and QC processed using Metabolon's hardware and software.

Beginning with the OrigScale values from Metabolon, which are normalized in terms of raw area counts, we calculated total ion count in the outflow to visualize sample-wise variance. Next, we normalized each sample by its total ion count and rescaled each metabolite's value by dividing each metabolite by its root mean square. We visualized metabolite abundance using a heat map. Statistical analysis and heat map generation was performed using R (R Foundation for Statistical Computing). We performed differential expression analysis by using the limma[61] R package (v3.32.10) to fit a linear model to the data. $\log_2$ fold change, $P$ value and adjusted $P$ value were calculated for each comparison using an unmoderated Student's $t$-test and the false discovery rate (FDR) method for multiple testing correction[62]. Adjusted $P$ values are shown in volcano plots. To assess transporter mediated uptake and transfer from the luminal upper channel to the lower channel, glycyl-sarcosine (Gly-Sar, 1 mM, Sigma) alone or in combination with the specific PEPT1 transporter inhibitor Gly-Gly dipeptide (1 mM, Sigma) was added to the luminal medium and their abundance in the lower channel outflow was assessed.

**Statistical analysis.** Each intestine chip was used as a biological repeat for one terminal assay. Either a Student's $t$-test or 2-way analysis of variance (ANOVA) was performed to determine statistical significance, as indicated in the figure legends (error bars indicate s.e.m.; $P < 0.05$ was considered significant).

**Reporting summary.** Further information on research design is available in the Nature Research Reporting Summary linked to this article.

## Data availability

The main data supporting the results in this study are available within the paper and its Supplementary Information. The organ-chip microarray data are available from the Gene Expression Omnibus (GEO) database, under accession number GSE202282. The clinical mRNA-seq data referenced as a comparison are available from GEO under accession number GSE159495. The metabolomics data are available from GitHub at https://github.com/ranikay/eed-metabolomics-analysis. Source data are provided with this paper.

## Code availability

Custom code for the analysis of the metabolomics data is available at https://github.com/ranikay/eed-metabolomics-analysis.

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

## Acknowledgements

This research was sponsored by funding from the Bill and Melinda Gates Foundation (independent support to D.E.I), NIH award DK119488 (to D.T.B) and the Anne London Fellowship (to C.W.F), and the Wyss Institute for Biologically Inspired Engineering. This work was conducted with the support of a KL2 award (an appointed KL2 award to C.W.F.) from Harvard Catalyst/The Harvard Clinical and Translational Science Center (National Center for Advancing Translational Sciences, National Institutes of Health Award KL2 TR002542). The content is solely the responsibility of the authors and does not necessarily represent the official views of Harvard Catalyst, Harvard University and its affiliated academic healthcare centres, or the National Institutes of Health.

## Author contributions

A.B., C.W.F., G.G. and D.E.I. designed the research. A.B., C.W.F., B.S., W.C., A.N., N.L., S.S., S.K. and S.J.-F. performed experiments. A.B., C.W.F., W.C., R.K.P., D.M.C., A.P., J.G., R.P.-B., G.G. and D.E.I. analysed and interpreted the data. D.T.B. established and prepared human healthy organoids. J.I. and A.A. established and prepared human EED organoids. L.A.D. and S.R.M. provided the clinical data. A.B., C.W.F. and D.E.I. wrote the Article with input from G.G. All authors reviewed, discussed and edited the manuscript.

## Competing interests

D.E.I. holds equity in Emulate, Inc., chairs its scientific advisory board and is a member of its board of directors. The other authors declare no competing interests.

## Additional information

**Extended data** is available for this paper at https://doi.org/10.1038/s41551-022-00899-x.

**Correspondence and requests for materials** should be addressed to Donald E. Ingber.

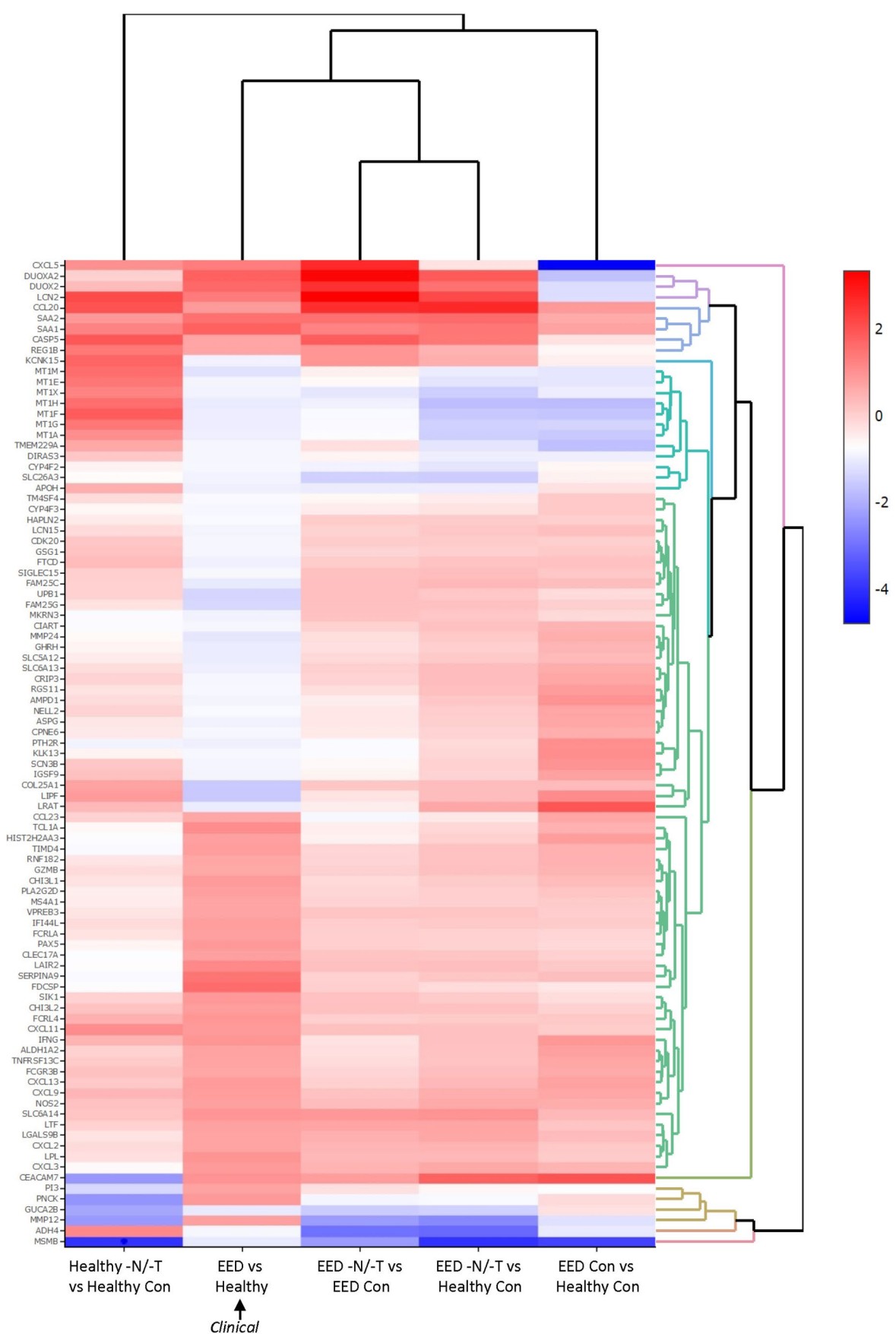

**Extended Data Fig. 1 | See next page for caption.**

**Extended Data Fig. 1 | Comparison of gene expression from the clinical EED signatures, for healthy and EED chips.** A comparison of the most upregulated and downregulated genes from the clinical EED signature for healthy and EED chips. Gene expression is depicted as a heatmap (red = upregulation, blue = downregulation), showing the expression pattern for EED chips in -N/-T media vs healthy chips in control media. 3 chips for each condition. Each chip corresponds to one biological replicate.

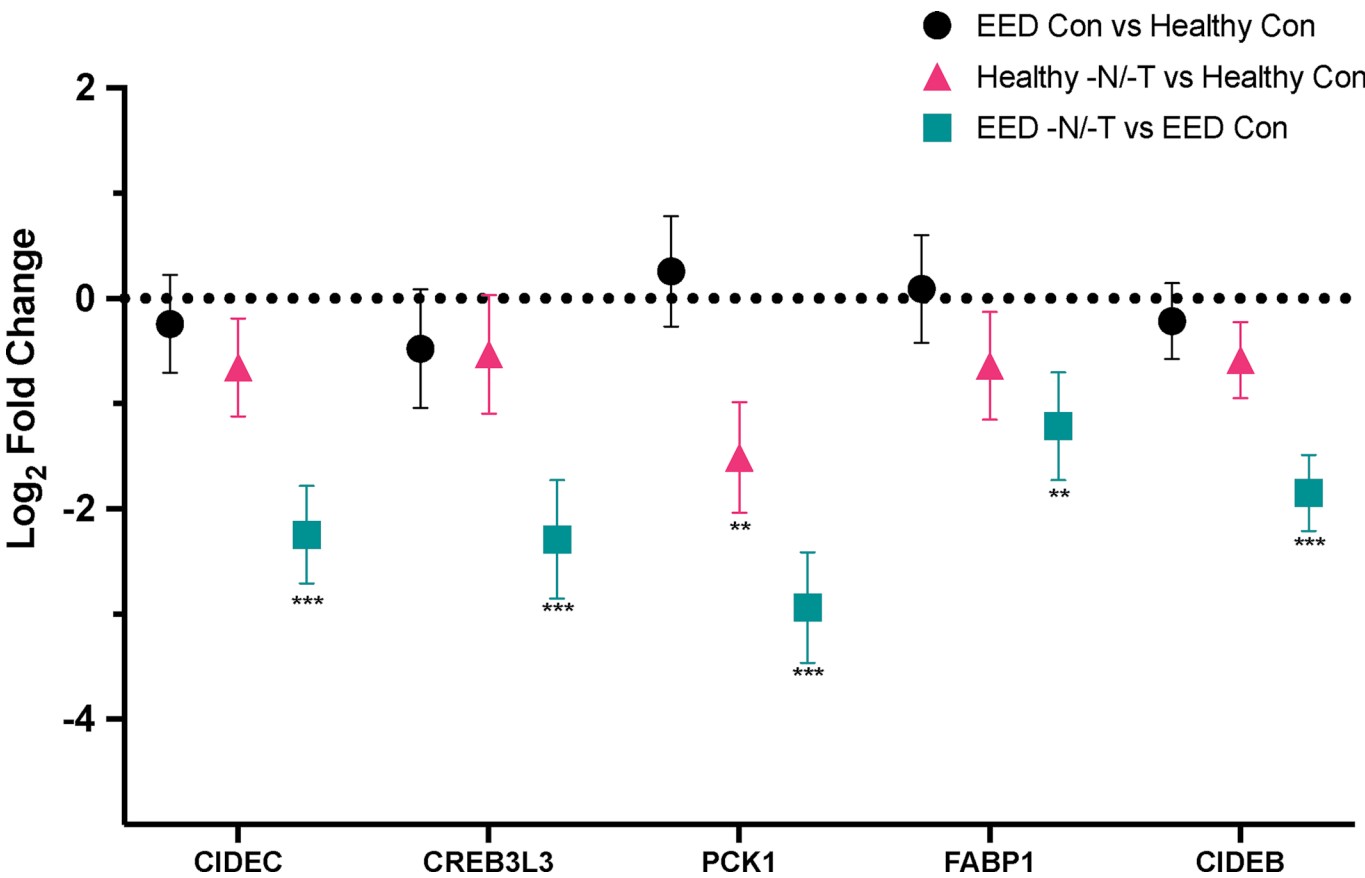

**Extended Data Fig. 2 | Gene expression of genes related to fatty acid absorption and processing.** Transcriptional pathway analysis revealed a strong theme of downregulation for genes related to fatty acid absorption and processing when EED chips were exposed to -N/-T media. This included a 4.7-fold downregulation of CIDEC ($q < 0.001$), a 4.9-fold downregulation of CREB3L3 ($q < 0.001$) and a 7.7-fold downregulation of PCK1 ($q < 0.001$), which are shown as the log2 fold change between the average expression levels along with the associated 95% confidence interval. 3 chips for each condition. Each chip corresponds to one biological replicate. \*\*, $p < 0.01$; \*\*\*, $p < 0.001$.

**a**

## REG3A

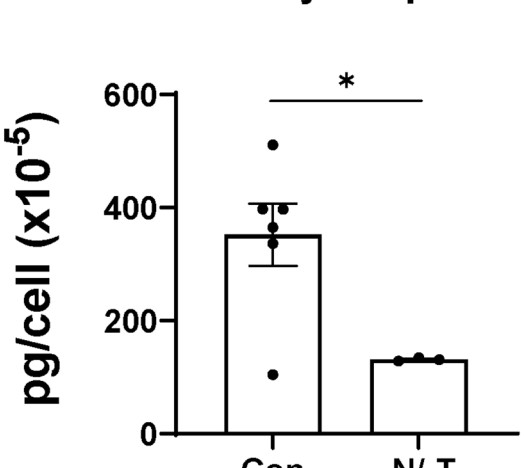

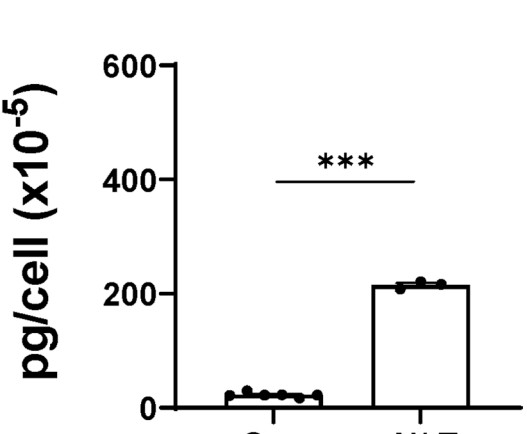

**b**

## LCN2

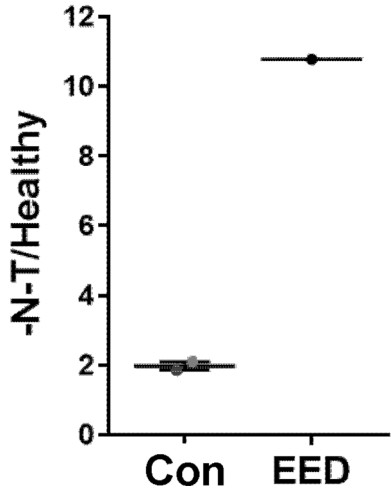

**Extended Data Fig. 3 | See next page for caption.**

**Extended Data Fig. 3 | Secreted REG3A and LCN2 levels measured in intestine-chip outflows. a**, Secreted REG3A levels measured in Intestine chips luminal outflows. healthy chip, $p = 0.029$, EED chip, $p < 0.000001$. n = 3 to 6 for each condition. Each chip corresponds to one biological replicate. **b**, Lipocalin 2 (LCN2) relative expression measured by ELISA. n = 2 for control and n = 1 for EED. Each chip corresponds to one biological replicate.

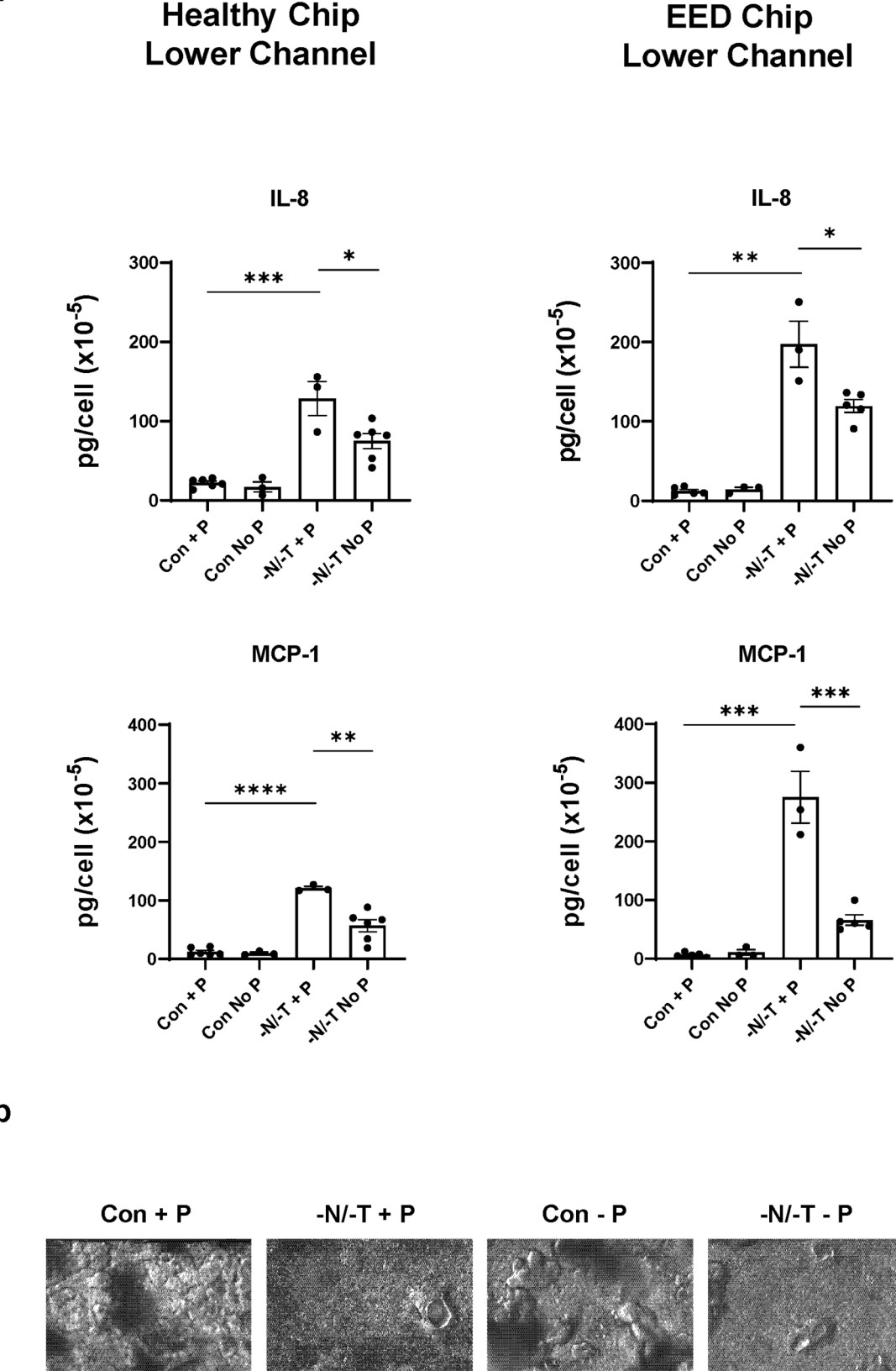

**a**

**Healthy Chip Lower Channel**

**EED Chip Lower Channel**

**Extended Data Fig. 4 | See next page for caption.**

**Extended Data Fig. 4 | Secreted IL-8 and MCP-1 levels measured in intestine-chip outflows. a**, Secreted IL-8 levels measured in the Intestine chips lower channel outflows, with and without peristalsis (P). Secreted MCP-1 levels measured in the Intestine chips lower channel outflows, with and without peristalsis. *, $p < 0.03$; **, $p < 0.005$; ***, $p < 0.0002$; ****, $p < 0.000002$. 3–6 chips for each condition. Each chip = One biological replicate. **b**, Differential Interference Contrast (DIC) imaging of Intestine chips, top-down view. Representative images of EED chips.

# Reporting Summary

## Statistics

For all statistical analyses, confirm that the following items are present in the figure legend, table legend, main text, or Methods section.

| n/a | Confirmed | |
|---|---|---|
| ☐ | ☒ | The exact sample size (*n*) for each experimental group/condition, given as a discrete number and unit of measurement |
| ☐ | ☒ | A statement on whether measurements were taken from distinct samples or whether the same sample was measured repeatedly |
| ☐ | ☒ | The statistical test(s) used AND whether they are one- or two-sided *Only common tests should be described solely by name; describe more complex techniques in the Methods section.* |
| ☐ | ☒ | A description of all covariates tested |
| ☐ | ☒ | A description of any assumptions or corrections, such as tests of normality and adjustment for multiple comparisons |
| ☐ | ☒ | A full description of the statistical parameters including central tendency (e.g. means) or other basic estimates (e.g. regression coefficient) AND variation (e.g. standard deviation) or associated estimates of uncertainty (e.g. confidence intervals) |
| ☐ | ☒ | For null hypothesis testing, the test statistic (e.g. *F*, *t*, *r*) with confidence intervals, effect sizes, degrees of freedom and *P* value noted *Give P values as exact values whenever suitable.* |
| ☒ | ☐ | For Bayesian analysis, information on the choice of priors and Markov chain Monte Carlo settings |
| ☒ | ☐ | For hierarchical and complex designs, identification of the appropriate level for tests and full reporting of outcomes |
| ☒ | ☐ | Estimates of effect sizes (e.g. Cohen's *d*, Pearson's *r*), indicating how they were calculated |

*Our web collection on statistics for biologists contains articles on many of the points above.*

## Software and code

Policy information about availability of computer code

| Data collection | Microsoft Excel, PRISM-ImageJ (FIJI), Metabolon. |
|---|---|
| Data analysis | IMARIS (IMARIS 7.6 F1 workstation; Bitplane Scientific Software)<br>ImageJ (FIJI)<br>Graphpad Prism software<br>R/R Studio: A language and environment for statistical computing (packages: SCAN.UPC, limma, heatmaply)<br>COmprehensive Multi-omics Platform for Biological InterpretatiOn (COMPBIO)<br>Metabolon Visual PHIL |

For manuscripts utilizing custom algorithms or software that are central to the research but not yet described in published literature, software must be made available to editors and reviewers. We strongly encourage code deposition in a community repository (e.g. GitHub). See the Nature Portfolio guidelines for submitting code & software for further information.

## Data

Policy information about availability of data

All manuscripts must include a data availability statement. This statement should provide the following information, where applicable:
- Accession codes, unique identifiers, or web links for publicly available datasets
- A description of any restrictions on data availability
- For clinical datasets or third party data, please ensure that the statement adheres to our policy

The main data supporting the results in this study are available within the paper and its Supplementary Information. The organ-chip microarray data are available

# Field-specific reporting

Please select the one below that is the best fit for your research. If you are not sure, read the appropriate sections before making your selection.

☒ Life sciences  ☐ Behavioural & social sciences  ☐ Ecological, evolutionary & environmental sciences

For a reference copy of the document with all sections, see nature.com/documents/nr-reporting-summary-flat.pdf

# Life sciences study design

All studies must disclose on these points even when the disclosure is negative.

| | |
|---|---|
| Sample size | Initial microarray experiments were carried out using one healthy donor (3 biological replicates) and one EED donor (3 biological replicates), and were reflective of a recent clinical EED transcriptomic signature derived from a larger population (SEEM study, with 25 healthy donors and 52 EED donors). Subsequent validation studies were carried out using 1–3 healthy donors (3–9 biological replicates) and 1–2 EED donors (3–8 biological replicates) per experiment (details provided in Methods and in figure legends). Sample sizes were determined on the basis of previous experimental experience as well as of the power of the statistical test performed. |
| Data exclusions | No data were excluded from the analyses. |
| Replication | Microarray sequencing data and metabolomics were done once with 3 biological replicates. All experiments were reproducible, as assessed in multiple intestine-chip culture devices per experiment. We did not have cases of irreproducibility. |
| Randomization | The intestine chips were randomly allocated into the experimental groups. |
| Blinding | The investigators were blinded to group allocation, as the chips were in random order when numbered at the time of seeding and placed in groups based on chip number. For other data collection and analysis, the investigators were not blinded to the identities of the samples. However, all experimental and control samples were collected and analysed at the same time under the same conditions, and the quantification was carried out with the same software settings. |

# Reporting for specific materials, systems and methods

We require information from authors about some types of materials, experimental systems and methods used in many studies. Here, indicate whether each material, system or method listed is relevant to your study. If you are not sure if a list item applies to your research, read the appropriate section before selecting a response.

### Materials & experimental systems

| n/a | Involved in the study |
|---|---|
| ☐ | ☒ Antibodies |
| ☒ | ☐ Eukaryotic cell lines |
| ☒ | ☐ Palaeontology and archaeology |
| ☒ | ☐ Animals and other organisms |
| ☐ | ☒ Human research participants |
| ☒ | ☐ Clinical data |
| ☒ | ☐ Dual use research of concern |

### Methods

| n/a | Involved in the study |
|---|---|
| ☒ | ☐ ChIP-seq |
| ☒ | ☐ Flow cytometry |
| ☒ | ☐ MRI-based neuroimaging |

## Antibodies

| | |
|---|---|
| Antibodies used | Primary antibody: Rabbit polyclonal to Apolipoprotein B (1:100, Abcam ab20737).<br><br>Secondary antibody: Donkey anti-Rabbit IgG (H+L) Cross-Adsorbed Secondary Antibody, DyLight 488 (1:1500; ThermoFisher SA5-10038)<br><br>Other labelling: Alexa Fluor 568 Phalloidin (Invitrogen A12380) and Wheat Germ Agglutinin, Alexa Fluor 488 Conjugate (ThermoFisher W11261) |
| Validation | Apolipoprotein B: recommended for immunofluorescence staining by the provider. We performed control experiments, including no primary antibody (negative) controls and comparison to experimental-group staining patterns. Additionally, this antibody has been cited in the literature many tens of times, as listed on the manufacturer's website. |

# Human research participants

Policy information about studies involving human research participants

| | |
|---|---|
| Population characteristics | Healthy cells isolated from human intestinal organoids were derived from biopsy specimens from the grossly uninflamed duodenum of pediatric patients (2–11 years old) diagnosed with ulcerative colitis during an endoscopy procedure.<br><br>EED cells isolated from human intestinal organoids were derived from biopsy specimens collected during an endoscopy procedure in malnourished pediatric patients (1.5–1.9 years old, male) diagnosed with EED. |
| Recruitment | Healthy duodenal biopsies were derived from biopsied tissue of a patient undergoing endoscopy for gastrointestinal complaints.<br><br>EED duodenal biopsies were derived from patients in the previously published Study of Environmental Enteropathy and Malnutrition (SEEM). The patients were recruited at birth and followed to 24 months of age. Duodenal tissue biopsies were collected during and endoscopy procedure in patients who failed to respond to nutritional intervention. |
| Ethics oversight | For the healthy duodenal biopsies, informed consent and developmentally appropriate assent were obtained at Boston Children's Hospital from the donors' guardian and the donor, respectively. All methods were carried out in accordance with the Institutional Review Board of Boston Children's Hospital (Protocol# IRB-P00000529) approval.<br><br>For the EED duodenal biopsies, informed consent was obtained at the housedhold level in a rural district of Matiari, Sind, Pakistan from the donors' guardians. All methods were carried out in accordance with the AKU Ethical Review Committee's approval (ERC number 3836-Ped-ERC-15). |

Note that full information on the approval of the study protocol must also be provided in the manuscript.

