## [Peer Review File · Nature Biomedical Engineering]

Nutritional deficiency in an intestine-on-a-chip recapitulates injury hallmarks associated with environmental enteric dysfunction

Corresponding author: Donald E. Ingber

Editorial note

This document includes relevant written communications between the manuscript's corresponding author and the editor and reviewers of the manuscript during peer review. It includes decision letters relaying any editorial points and peer-review reports, and the authors' replies to these (under 'Rebuttal' headings). The editorial decisions are signed by the manuscript's handling editor, yet the editorial team and ultimately the journal's Chief Editor share responsibility for all decisions.

Any relevant documents attached to the decision letters are referred to as **Appendix #**, and can be found appended to this document. Any information deemed confidential has been redacted or removed. Earlier versions of the manuscript are not published, yet the originally submitted version may be available as a preprint. Because of editorial edits and changes during peer review, the published title of the paper and the title mentioned in below correspondence may differ.

As noted in the correspondence below, the manuscript underwent two rounds of peer-review at *Nature Biomedical Engineering*, after being transferred from *Nature Medicine* after peer review. The peer-review information included in this document pertains to the post-transfer process.

Correspondence

Sat 22 Jan 2022

Decision on Article nBME-21-2905-T

Dear Dr Ingber,

Thank you for your revised manuscript, "Nutritional deficiency recapitulates intestinal injury associated with environmental enteric dysfunction in patient-derived Organ Chips".

As communicated in earlier e-mail correspondence, the revised manuscript has been reviewed by Reviewers #2 and #3 recruited by *Nature Medicine* and who assessed the previous version of the manuscript (Reviewer #1 declined to re-review), and by a new expert (Reviewer #4).

In the reports, which you will find at the end of this message, you will see that the reviewers acknowledge the improvements to the work and raise a few additional suggestions and technical criticisms that we hope you will be able to address.

When you are ready to resubmit your manuscript, please upload the revised files, a point-by-point rebuttal to the comments from all reviewers, and the reporting summary.

As a reminder, please follow the following recommendations:

* Clearly highlight any amendments to the text and figures to help the reviewers and editors find and understand the changes (yet keep in mind that excessive marking can hinder readability).- * If you and your co-authors disagree with a criticism, provide the arguments to the reviewer (optionally, indicate the relevant points in the cover letter).
- * If a criticism or suggestion is not addressed, please indicate so in the rebuttal to the reviewer comments and explain the reason(s).
- * Consider including responses to any criticisms raised by more than one reviewer at the beginning of the rebuttal, in a section addressed to all reviewers.
- * The rebuttal should include the reviewer comments in point-by-point format (please note that we provide all reviewers with the reports as they appear at the end of this message).
- * Provide the rebuttal to the reviewer comments and the cover letter as separate files.

We look forward to receive a further revised version of the work. Please do not hesitate to contact me should you have any questions.

Best wishes,

Pep

Pep Pàmies
Chief Editor, Nature Biomedical Engineering

Reviewer #2 (Report for the authors (Required)):

Summary

Amir Bein, et.al used Human Intestine Chip technology to create an in vitro environmental enteric dysfunction (EED) model that allows for the measurement of transport and absorption across enterocytes, unique from previous organoid models that use static ECM gels. This intestine chip model was used to investigate the effect of nutrient deficiency on the functional and phenotypic manifestation of EED using biopsies from 3 healthy (Boston) and 2 EED (Pakistan) donors. The great strength of this paper is that it combines multiple approaches (transcriptome, immunofluorescence microscopy, nutrient flux) to elucidate what is going wrong in EED, and the experimental approach offers, probably for the first time in human tissue, confirmation that specific nutritional deficiencies contribute to pathophysiology. The results yield remarkable and unique insights into the biology of intestinal responses to nutrient depletion.

Degree of advance

This is a considerable step forward, as it demonstrates the power of this new bioengineering approach.

Implications

The paper is fundamentally about demonstrating the validity and utility of this new model. However, the data also add to a substantial body of work now showing that derangements in specific nutrients and their metabolic pathways can contribute to environmental enteropathy. This has not been possible until now as causation has been very uncertain.

Technical criticisms

How were N and T removed from the culture and perfusion media? Could this have introduced any toxic moieties?

Missing citations

The authors have greatly improved the Discussion in terms of comparison with relevant transcriptomic literature. However, this reviewer still feels that they under-sell the close transcriptional profiles seen in the studies from Bangladesh, Pakistan and Zambia. Many of the DEGs are identical and could be spelled out. This will strengthen the case that this is a valid model.

Minor points

1 Supplementary Figure 5 now shows “Active Transport” on the y axis, in response to my comment that this is probably what is being measured. However, it has no units. This reviewer is also unsure that the authors have got the point that Active Transport can only be confirmed if it can be shown that solute movement is occurring against a concentration gradient. This term should only be used if that can be confirmed.

2 Supplementary Figure 6 has a truncated Color Key. Also, what is DM?

3 The abbreviation “WGA” is not explained. Presumably Wheat Germ Agglutinin?

Reviewer #3 (Report for the authors (Required)):

This report a novel method to describe the pathophysiology of environmental enteropathy in your children from Pakistan. The findings from this sample is compared with healthy controls from the US.

Thank you for clarifying some of the issues that were raised based on reading the the first submission.

The age of the children were 3 for 6 months when they were identified and they were defined as cases at 9 months. Was there only two children from Pakistan, and I could not find the exact age for when they were sampled. Similarly, the Boston-controls were between 2 and 11 years old and I assume that at least one child was 2 years and another 11 years old. I suggest that you provide the exact age of all five children who provided biological samples for the experiment. More importantly, the gut goes through a substantial development throughout the first months and years of life. A 11 year-old gut resembles more an adult gut than an infant gut. The difference in age distribution between the cases and the controls is hard to justify. The Pakistani children were identified at 3-6 months of age and sampled when they were 1-2 years. More details on how these two children were selected is needed. In the current version the populations are described both in lines 445-450 and lines 469-473 and does not offer much clarity regarding this issue.

More clarity regarding the children is still needed.

Reviewer #4 (Report for the authors (Required)):

In this manuscript, the authors established an in vitro environmental enteric dysfunction(EED) model using human intestine-on-a-chip comprising microfluidic device and intestinal organoid-derived epithelial cells from EED patients. Exposure of EED chips to nutritionally deficient condition with niacinamide- and tryptophan-deficient (-N/-T) medium recapitulated the key transcriptional signatures and phenotypic features of EED-associated intestinal injury including upregulation of intestinal injury-associated genes, severe villus blunting, barrier dysfunction, impaired fatty acid uptake and amino acid transport. Therefore, the developed human intestine chip with EED-associated intestinal injury would be able to provide a highly valuable preclinical model for the research of EED pathophysiology and EED drug testing. Given a recent high occurrence of EED from intestinal injury caused by perturbation of environmental factors, this study demonstrating the application of organ-on-a-chip technology and patient organoids to EED modeling is attractive and quite timely. The manuscript was concisely well written and provides meaningful data to support the effectiveness of the EED intestine chip. This reviewer would suggest several major and minor issues to be addressed for further improving the manuscript and clarifying some unclear issues before publication.

Major comments

1. Overall, the results in this study are very interesting and the study is novel because there has been no human in vitro EED model before. Current study has mainly focused on transcriptional and functional analyses to validate the EED intestine chip model, but this reviewer would like to recommend the authors to further characterize their EED chip model in terms of alteration in cellular compositions. EED definitely affects populations of organoid-derived cells and thus perturbation of intestinal cell types needs to be compared between EED intestine chip and healthy intestine chip with immunofluorescence staining and qPCR for major intestine cell markers. Further, cell composition change in EED intestine chip under nutrition-

deficient condition would be interesting data as well.

2. Besides additional characterization of EED intestine chip requested in my previous comment above, characterization data of EED patient-derived organoids should be provided as supplementary figures. As EED is influenced more by environmental factors than genetic predisposition, it is likely that EED organoids derived from biopsies of EED patients lose their pathological features during prolonged culture in vitro. Current manuscript does not contain any characterization data of EED organoid itself. Thus, it would be recommended to check whether the organoids derived from EED patients retain the features of EED even after multiple passages. Also, please clarify the passage number of the EED organoids used to make the EED chips. I think the authors already have the relevant data and information to validate EED phenotypes of EED intestine organoids before use for chip fabrication.

3. The sentence at line 166-167 is a description of a healthy -N/-T chip? Overall, the paragraph at line 158-172 explains the upregulated and downregulated pathway in the EED -N/-T chip, and the Figure 1e also describes the EED -N/-T chip. If this sentence really mentions a healthy -N/-T chip, the authors should provide the data to support this statement (e.g., the affected pathway in healthy -N/-T chip versus healthy control chip).

4. The authors demonstrated that the EED -N/-T chip has higher similarity with the clinical data of EED patients than the healthy -N/-T chip through Figure 1b-d and Supplementary Figures 1 and 2. However, in Figure 2e, why did the Papp value change more dramatically in the healthy chip than in the EED chip under the same -N/-T condition (8.9-fold increase in the healthy -N/-T chip and 2.5-fold increase in the EED -N/-T chip)? In addition, when comparing the Papp values of the healthy -N/-T chip and the EED -N/-T chip, it seems that the barrier function of the healthy -N/-T chip was worse than the EED -N/-T chip. Similarly, in Figure 2c comparing the villi height change in the control medium and the -N/-T medium condition, the villi height of the healthy chip decreased more significantly. According to the data in Figure 1, pathological change due to nutritional deficiency should be more pronounced in the EED chip. Thus, it is required to provide reasonable explanation for such inconsistency.

5. In this study, healthy intestine and EED intestine chips were prepared with organoids from tissue biopsies of 3 healthy and 2 EED patients. I think the number of tissue samples for organoid preparation may not be sufficient to support the robustness of intestine chips in terms of quality control. Thus, I wonder the degree of a variation between the intestine chips fabricated with different tissue batches (both healthy and EED chips) in transcriptome levels and physiological features. Principal component analysis (PCA) of transcriptome levels can help to check the variations of samples in each group as well as similarity and difference between the chip models.

6. The presence of immune cells is of great importance for precise disease modeling associated with intestinal injury. Actually, pathophysiology of EED is closely related to the activity of immune cells. Although here the authors also showed upregulation of key inflammatory markers and cytokines in the chip models exposed to nutritional deficiency, intestinal organoids used for chip fabrication did not contain immune cells, which may interfere precise EED modeling and subsequent drug testing for identifying EED therapeutics. In-depth discussion on this issue needs to be provided.

7. The study has reported successful establishment and validation of EED intestine chip models but lacks applications of the developed chips. In this context, I just wonder whether the authors have ever tried microbiome co-culture in EED intestine chips. I guess EED intestine and healthy intestine chips show quite different capability for microbiome culture. Even simple, preliminary tests to observe phenotypic and functional recovery of injured intestinal epithelium in EED chips by potential EED drugs and nutritional supplementation would also be able to significantly strengthen the impact of the manuscript.

Minor comments

1. In Figure 2d, quantification of the thinning of the mucous layer in intestine chips due to nutritional deficiencies would be required.

2. In Figure 3b, please add the representative images of the ApoB immunofluorescence staining. Likewise, for Supplementary Figure S8, it would be needed to include the images of epithelium in the chips with or without exposure to mechanical peristalsis-like deformation. I wonder whether peristalsis-like deformation induced morphological changes in villus-like structures as well as cytokine secretion profiles.

3. As mentioned in the Introduction, lactulose-mannitol (L:M) test has been most frequently used to confirm EED pathophysiology, which would be worth being added in the manuscript.
4. There are several typos throughout the manuscript. On line 96, please delete [REF]. On line 119, there is unnecessary parenthesis “)”.
5. In the order of figures, it is recommended to number the figures according to the order indicated in the text. For example, Supplementary Figure S5 and S6.
6. The description of p values is missing in the legend of Figure 4a. Please also specify the sample number in the legends of each figure as either biological replicates or technical replicates.
7. It is recommended to move the full name of CD36 on line 399 to its first appearance on line 214.

Sat 02 Apr 2022

Decision on Article nBME-21-2905A

Dear Dr Ingber,

Thank you for your revised manuscript, "Nutritional deficiency recapitulates intestinal injury associated with environmental enteric dysfunction in patient-derived Organ Chips". Having consulted with the reviewers, I am pleased to write that we shall be happy to publish the manuscript in *Nature Biomedical Engineering*, provided that the points specified in the attached instructions file are addressed.

When you are ready to submit the final version of your manuscript, please upload the files specified in the instructions file.

Best wishes,

Pep

Pep Pàmies
Chief Editor, Nature Biomedical Engineering

P.S. Nature Research journals encourage authors to share their step-by-step experimental protocols on a protocol-sharing platform of their choice. Nature Research's Protocol Exchange is a free-to-use and open resource for protocols; protocols deposited in Protocol Exchange are citable and can be linked from the published article. More details can be found at www.nature.com/protocolexchange/about.

Reviewer #2 (Report for the authors (Required)):

The authors have fully addressed by comments and suggestions

Reviewer #3 (Report for the authors (Required)):

The authors have responded well to our comments and criticism. No further comments from me.

Reviewer #4 (Report for the authors (Required)):

The authors have properly addressed all issues raised by this reviewer and the issues have now been mostly resolved. The rebuttal provided in the letter are also reasonable. Accordingly, I think this revised version of the manuscript is now acceptable for publication.

Nature Biomedical Engineering is a Transformative Journal. Authors may publish their research with us through the traditional subscription access route, or make their paper immediately open access through payment of an article-processing charge. More information about publication options is available.

You may need to take specific actions to comply with funder and institutional open-access mandates. If the work described in the accepted manuscript is supported by a funder that requires immediate open access (as outlined, for example, by Plan S) and your manuscript was originally submitted on or after January 1st 2021, then you will need to select the gold OA route. Authors selecting subscription publication will need to accept our standard licensing terms (including our self-archiving policies), and these will supersede any other terms that the author or any third party may assert apply to any version of the manuscript.

Rebuttal 1

RESPONSE TO REVIEWERS

Bein et al. (nBME-21-2905-T)

REVIEWER #2:

1. *How were N and T removed from the culture and perfusion media? Could this have introduced any toxic moieties?*

The deficiency in N and T was created by using a custom made base medium DMEM-F12 (Thermo Fisher), where these nutrients were not added during production.

2. *The authors have greatly improved the Discussion in terms of comparison with relevant transcriptomic literature. However, this reviewer still feels that they under-sell the close transcriptional profiles seen in the studies from Bangladesh, Pakistan and Zambia. Many of the DEGs are identical and could be spelled out. This will strengthen the case that this is a valid model.*

We agree that the favorable comparison of transcriptional changes in EED Chips exposed to nutritional deficiency to those seen in clinical EED is an exciting and central finding of this work. We have now spelled out specific examples of overlapping DEGs (line 342-345), including reference to the studies suggested.

3. *Supplementary Figure 5 now shows “Active Transport” on the y axis, in response to my comment that this is probably what is being measured. However, it has no units. This reviewer is also unsure that the authors have got the point that Active Transport can only be confirmed if it can be shown that solute movement is occurring against a concentration gradient. This term should only be used if that can be confirmed.*

In this assay, we used a specific inhibitor to confirm that the transfer of Gly-Sar is through an intracellular path and not a simple intercellular diffusion. We agree that the term “Active Transport” is appropriate when the movement of the solute is against the concentration gradient. Thus, we have revised and added units to Y axis title: “Transport (cm/s)”

4. *Supplementary Figure 6 has a truncated Color Key. Also, what is DM?*

We have corrected the truncated text of the Color Key. DM = differentiation medium. We have added this definition to Supplementary Figure 6 legend as well as to Figure 4 legend.

5. *The abbreviation “WGA” is not explained. Presumably Wheat Germ Agglutinin?*

We now define Wheat Germ Agglutinin (WGA) at the first appearance in the text line 502.

REVIEWER #3:

1. *The age of the children were 3 for 6 months when they were identified and they were defined as cases at 9 months. Was there only two children from Pakistan, and I could not find the exact age for when they were sampled. Similarly, the Boston-controls were between 2 and*

11 years old and I assume that at least one child was 2 years and another 11 years old the, I suggest that you provide the exact age of all five children who provided biological samples for the experiment.

The 2 EED donors from Pakistan were sampled at the age of 1.5 and 1.9 years old. We now specify the age of all donors in the text (lines 424 and 431) in addition to the detailed description in Supplementary Table 3.

2. More importantly, the gut goes through a substantial development throughout the first months and years of life. A 11 year-old gut resembles more an adult gut than an infant gut. The difference in age distribution between the cases and the controls is hard to justify. The Pakistani children were identified at 3-6 months of age and sampled when they were 1-2 years. More details on how these two children were selected is needed. In the current version the populations are described both in lines 445-450 and lines 469-473 and does not offer much clarity regarding this issue. More clarity regarding the children is still needed.

The intestine undergoes maturation with age and indeed, the length of the intestine changes in children over time (PMID: 28631339). We agree that there are differences between the intestine in a 2 versus an 11 year old, yet there are obvious limitations to the availability of donor derived samples. Moreover, the case here is different as we are using primary cells that are seeded in a synthetic system to rebuild the intestinal architecture over a period of two weeks. The architecture of the Intestine chip is constructed and undergoes differentiation over uniform length of time for all donors compared, thereby key features such villi development, junction formation and differentiation into intestinal cell types is taking place simultaneously on all chips. Most importantly, we used a 2 year old along with older donors in the control group and show similar key physiological characteristics (e.g., barrier function, villi formation) within chips created with cells from both sources. Thus, we do not believe this is a significant limitation in this study.

REVIEWER #4:

1. Current study has mainly focused on transcriptional and functional analyses to validate the EED intestine chip model, but this reviewer would like to recommend the authors to further characterize their EED chip model in terms of alteration in cellular compositions. EED definitely affects populations of organoid-derived cells and thus perturbation of intestinal cell types needs to be compared between EED intestine chip and healthy intestine chip with immunofluorescence staining and qPCR for major intestine cell markers. Further, cell composition change in EED intestine chip under nutrition-deficient condition would be interesting data as well.

To address the Reviewer's concerns about changes in cellular composition, we now include an analysis comparing gene expression changes in our model with previously identified clusters of intestinal cell-type markers (Supplementary Figure 3 and Supplementary Table 2). We find that EED Chips express lower levels of enterocyte markers when compared to Healthy Chips in control medium, and further downregulate these markers upon exposure to nutritional deficiency. Paneth cell markers are similarly decreased in EED Chips compared to Healthy Chips in control medium; however, upon

exposure to nutritional deficiency, Paneth cell markers are upregulated in Healthy Chips but downregulated in EED Chips. These findings are consistent with the histopathological finding of Paneth cell depletion in EED patient intestinal biopsies. We thank the Reviewer for this helpful suggestion as these new data strengthen the manuscript.

2. Besides additional characterization of EED intestine chip requested in my previous comment above, characterization data of EED patient-derived organoids should be provided as supplementary figures. As EED is influenced more by environmental factors than genetic predisposition, it is likely that EED organoids derived from biopsies of EED patients lose their pathological features during prolonged culture in vitro. Current manuscript does not contain any characterization data of EED organoid itself. Thus, it would be recommended to check whether the organoids derived from EED patients retain the features of EED even after multiple passages. Also, please clarify the passage number of the EED organoids used to make the EED chips. I think the authors already have the relevant data and information to validate EED phenotypes of EED intestine organoids before use for chip fabrication.

In response to this concern, we now describe the range of passage numbers for the organoids used in our studies in Supplementary Table 3. Our data support your larger point that crucial features of EED can be lost with prolonged culture. However, in previous studies, we demonstrated that when cells from duodenal organoids are seeded and cultured on chip in the presence of appropriate mechanical cues, the intestinal epithelium that differentiates on-chip is more transcriptionally similar to in vivo duodenum than the cultured organoids from which the cells were derived (PMID: 29440725). Intestinal chips seeded with EED donors that are grown in control medium (EED Con) have only limited transcriptional similarity to the clinical EED transcriptional signature. This is likely due to a lack of environmental factors and in fact, when these chips are exposed to the environmental stimulus of malnutrition (EED -N/-T), they undergo many transcriptional changes that directly align with the clinical EED transcriptional signature. This is partially true for chips seeded with enteroids from healthy donors (Healthy Con vs Healthy -N/-T), but the specific combination of EED patient intestinal epithelium exposed to nutritional deficiency exhibits transcriptional changes that most closely resemble those seen in biopsies from actual EED patients requires the. This suggest that, at least in the nutritionally refractory patients studied here, that a combination of genetic/epigenetic changes and environmental factors underlies pathophysiology.

3. The sentence at line 166-167 is a description of a healthy -N/-T chip? Overall, the paragraph at line 158-172 explains the upregulated and downregulated pathway in the EED -N/-T chip, and the Figure 1e also describes the EED -N/-T chip. If this sentence really mentions a healthy -N/-T chip, the authors should provide the data to support this statement (e.g., the affected pathway in healthy -N/-T chip versus healthy control chip).

This sentence should refer to genes rather than pathways. We have corrected the text and now include a list of the affected genes (**ATF4, TP53, AARS, YARS, MDM2, CCND2** and several amino acid transporters) with a reference to Supplementary Table 1 where these gene changes are enumerated.

4. The authors demonstrated that the EED -N/-T chip has higher similarity with the clinical data of EED patients than the healthy -N/-T chip through Figure 1b-d and Supplementary Figures 1 and 2. However, in Figure 2e, why did the Papp value change more dramatically in the healthy chip than in the EED chip under the same -N/-T condition (8.9-fold increase in the healthy -N/-T chip and 2.5-fold increase in the EED -N/-T chip)? In addition, when comparing the Papp values of the healthy -N/-T chip and the EED -N/-T chip, it seems that the barrier function of the healthy -N/-T chip was worse than the EED -N/-T chip. Similarly, in Figure 2c comparing the villi height change in the control medium and the -N/-T medium condition, the villi height of the healthy chip decreased more significantly. According to the data in Figure 1, pathological change due to nutritional deficiency should be more pronounced in the EED chip. Thus, it is required to provide reasonable explanation for such inconsistency.

It is likely that there are adaptive features of EED intestinal epithelium that develop in the setting of constant exposure to environmental factors. One previously identified adaptation is an upregulation in Wnt signaling during protein deprivation (PMID: 30630118). Our top upregulated gene in EED Chips exposed to nutritional deficiency was the stem cell marker SMOC2 and we see a trend toward upregulation of LGR5 (3.6 fold, $p=0.12$). It is tempting to speculate that, similar to the previous literature, this may represent a compensatory adaptation that is absent in healthy intestinal epithelium upon exposure to nutritional deficiency. For example, one possibility is that compensatory changes in proliferative stem cells in EED Chips could increase epithelial cell densities in the epithelium and thus, enhance barrier integrity relative to healthy chips. We now cover these points in the Discussion.

5. In this study, healthy intestine and EED intestine chips were prepared with organoids from tissue biopsies of 3 healthy and 2 EED patients. I think the number of tissue samples for organoid preparation may not be sufficient to support the robustness of intestine chips in terms of quality control. Thus, I wonder the degree of a variation between the intestine chips fabricated with different tissue batches (both healthy and EED chips) in transcriptome levels and physiological features. Principal component analysis (PCA) of transcriptome levels can help to check the variations of samples in each group as well as similarity and difference between the chip models.

As requested, we added Supplementary Figure 2 that includes a PCA plot of the transcriptome level analyses, which demonstrates uniformity of chips within the different experimental groups. We also added text to describe this finding in the Results.

6. The presence of immune cells is of great importance for precise disease modeling associated with intestinal injury. Actually, pathophysiology of EED is closely related to the activity of immune cells. Although here the authors also showed upregulation of key inflammatory markers and cytokines in the chip models exposed to nutritional deficiency, intestinal organoids used for chip fabrication did not contain immune cells, which may interfere precise EED modeling and subsequent drug testing for identifying EED therapeutics. In-depth discussion on this issue needs to be provided.

We agree that incorporation of immune cells would contribute and further increase the physiological accuracy of the EED Intestine Chip model. However the development of such complexity is beyond the scope of the current study, which focuses

on the ability of this Organ Chip system to model many (but not all) features of EED exhibited in patients. We have now added language to cover this point in the Discussion.

7. The study has reported successful establishment and validation of EED intestine chip models but lacks applications of the developed chips. In this context, I just wonder whether the authors have ever tried microbiome co-culture in EED intestine chips. I guess EED intestine and healthy intestine chips show quite different capability for microbiome culture. Even simple, preliminary tests to observe phenotypic and functional recovery of injured intestinal epithelium in EED chips by potential EED drugs and nutritional supplementation would also be able to significantly strengthen the impact of the manuscript.

As described in the response to the last comment, these are precisely the type of studies that can be carried out in the future now that this model exists, but this is far beyond the scope of the present study that focuses on demonstration of the basic creation and initial validation of these model.

8. In Figure 2d, quantification of the thinning of the mucous layer in intestine chips due to nutritional deficiencies would be required.

We now added quantification of the mucus thickness in Fig, 2f) and revised the text to describe the results the correct order of appearance.

9. In Figure 3b, please add the representative images of the ApoB immunofluorescence staining. Likewise, for Supplementary Figure S8, it would be needed to include the images of epithelium in the chips with or without exposure to mechanical peristalsis-like deformation. I wonder whether peristalsis-like deformation induced morphological changes in villus-like structures as well as cytokine secretion profiles.

We now include representative images of the ApoB immunofluorescence staining in Supplementary Figure 6. We also added images of the epithelium with and without peristalsis like deformations in Supplementary Figure 11b. We have previously published work showing that dynamic fluid flow is the most critical requirement for villus differentiation in Intestine Chips, although peristalsis-like motions can also increase the efficiency of some differentiation characteristics as well as the response to commensal microbiome (PMID: 22434367; PMID: 23817533, PMID: 29440725, PMID: 26668389; PMID: 31778828), and we now cite relevant publications.

10. As mentioned in the Introduction, lactulose-mannitol (L:M) test has been most frequently used to confirm EED pathophysiology, which would be worth being added in the manuscript.

We agree that the L:M could be an interesting comparison, however, this protocol has never been validated using Organ Chip technology. In contrast, quantification of Papp using tracer molecules as we did in this study is standard practice in this field. To assess nutrient uptake and transfer we used a more complex method and analyzed metabolite concentrations in the lumen and basal compartments. It is important to note that metabolite concentrations have been previously shown to correlate with L:M measurements (<https://www.nature.com/articles/srep28009>

11. *There are several typos throughout the manuscript. On line 96, please delete [REF]. On line 119, there is unnecessary parenthesis “)”.*

We have corrected the typos in the text.

12. *In the order of figures, it is recommended to number the figures according to the order indicated in the text. For example, Supplementary Figure S5 and S6.*

We have corrected the numbering of figures according to order of appearance in the text

13. *The description of p values is missing in the legend of Figure 4a. Please also specify the sample number in the legends of each figure as either biological replicates or technical replicates.*

A p-value adjusted for multiple tests by the Benjamini-Hochberg procedure (also FDR value) is mentioned for the relevant genes in legend to Fig. 4a. We now specify the sample number as chips (biological) or technical replicates in the legend of each figure.

14. *It is recommended to move the full name of CD36 on line 399 to its first appearance on line 214.*

We moved the full name of CD36 to line 214.